# A digital platform with activity tracking for energy management support in long COVID: a randomised controlled trial

Nilihan EM Sanal-Hayes ®[1], Lawrence D. Hayes ®[2] ✉, Jacqueline L. Mair ®[3,4,5], Antonio Dello Iacono ®[6], Joanne Ingram ®[7], Marie Mclaughlin ®[8], Jane Ormerod[9], David Carless ®[6], Natalie Hilliard[10], Rachel Meach ®[11] & Nicholas F. Sculthorpe ®[6] ✉

In a 6-month pragmatic randomised controlled trial (RCT; ISRCTN16033549), we compared a just-in-time intervention to support energy management in adults with long COVID (LC) to standard care. Participants received either the 'Pace Me' app and a wearable activity tracker (intervention) or an app only with data entry screens (control). The intervention group received just-in-time messages on energy management when they reached 50%, 75%, and 100% of their daily 'activity allowance'. The primary outcome was post-exertional malaise (PEM) measured by the DePaul Symptom Questionnaire-PEM (DSQ-PEM). Of 369 participants assessed for eligibility, 250 participants were randomised 1:1, and 77 controls and 84 intervention participants were included in the final per-protocol analysis. There was no time by group interaction for the DSQ-PEM. The intervention group value was 48 (95% CI 44-53) at baseline and 46 (95% CI 41-51) post-intervention (arbitrary units). The control group value was 47 (95% CI 42-52) at baseline and 44 (95% CI 39-49) at follow-up (interaction effect p = 0.614, $\eta^2$p = 0.002; trivial). No individual question exhibited an interaction effect (p > 0.05). Although the intervention had minimal effect compared to control, the substantial recovery rates previously reported in LC, coupled with our wide inclusion criteria may have masked intervention effects. Therefore, future studies should consider this energy management framework in conditions without such recovery rates, such as CFS.

Following the UK outbreak of the Sars-Cov-2 (COVID-19) virus in March 2020, a significant proportion of those who became infected experienced persistent symptoms lasting several months commonly known as long COVID (LC). Symptoms of LC include pain, lethargy, exercise intolerance, myalgia, cognitive impairment, and fatigue[1,2]. Symptoms can be disabling even if initial effects were mild, and can occur immediately after infection or some weeks later[3]. These accounts appear similar to post-exertional malaise (PEM; described as an

[1]School of Health and Society, University of Salford, Salford, UK. [2]Lancaster Medical School, Lancaster University, Lancaster, UK. [3]Future Health Technologies, Singapore-ETH Centre, Campus for Research Excellence and Technological Enterprise (CREATE), Singapore, Singapore. [4]Yong Loo Lin School of Medicine, National University of Singapore, Singapore, Singapore. [5]Department of Management, Technology and Economics, Centre for Digital Health Interventions, ETH Zurich, Zurich, Switzerland. [6]Sport and Physical Activity Research Institute, School of Health and Life Sciences, University of the West of Scotland, Glasgow, UK. [7]School of Education and Social Sciences, University of the West of Scotland, Glasgow, UK. [8]Physical Activity for Health Research Centre, Institute for Sport, P.E. and Health Sciences, University of Edinburgh, Moray House School of Education and Sport, Edinburgh, UK. [9]Long-COVID Scotland, Aberdeen, UK. [10]Physios for ME, London, UK. [11]Department of Earth Sciences, Durham University, Durham, UK. ✉e-mail: L.Hayes4@lancaster.ac.uk; nicholas.sculthorpe@uws.ac.uk

intolerance to mental and physical exertion, triggering an aggravation of symptoms typically lasting >14 h up to several days[4]) reported by people with conditions, such as myalgic encephalomyelitis/chronic fatigue syndrome (ME/CFS).

LC is associated with highly variable symptom loads, fluctuating between periods of relative alleviation and exacerbation, often preceded by exertion[5]. Our public patient involvement (PPI) group and others[6] describe 'push-crash' cycles where a perception of getting better leads to a push to return to normal activities precipitating a subsequent crash with worsening of symptoms, similar to PEM. Early work confirmed that 50% of people with LC meet the technical definition of PEM lasting >14 h, while nearly all report some PEM-like symptoms lasting up to 14 h[6,7].

The cause of persistent symptoms in LC is poorly understood, and a pathophysiological mechanism remains elusive. This is particularly true for those whose initial symptoms were mild and for non-hospitalised individuals[5], who did not receive any treatment that might explain prolonged rehabilitation (e.g. mechanical ventilation). Without a clear underlying mechanism, treatment options have focused on symptom management to help people with LC manage activities of daily living. Building on the symptom similarities between LC and ME/CFS, the most common symptom management strategy in LC is energy management, the only strategy currently recommended for ME/CFS[8]. Energy management uses techniques such as planning, feedback, and activity diaries to help individuals manage their 'energy envelope' to limit the debilitating consequences of their condition by reducing or avoiding periods of symptom exacerbation[9–11].

Energy management is suitable for self-management and is associated with higher adherence and greater patient-reported benefit[12]. However, despite multiple underlying theories[9,13], concrete steps for symptom self-management are rare. Energy management also requires participants to recall and compare two poorly described constructs; 'energy availability' and 'energy use', which our PPI group and evidence from the ME/CFS literature find challenging, especially when dealing with impaired cognition and memory[14]. Additionally, the dynamic nature of energy management requires recalling activities that might have triggered previous bouts of PEM-like symptoms, even if they occurred weeks or months earlier. Despite its widespread use (in ME/CFS), evidence to support the efficacy of energy management in preventing bouts of PEM is extremely limited[11,15]. Encouragingly, Parker and colleagues[16] provided some preliminary evidence that a structured energy management protocol resulted in a significant reduction in the number of PEM episodes in people with LC. However, this intervention seems analogous to graded exercise therapy (a gradual increase in physical activity over time), rather than energy management per se, which is highly contentious with the LC and ME/CFS support groups. To our knowledge, there are no contemporary efficacy trials that leverage mobile and wearable technology, which could address many of the challenges associated with energy management[11,15]. A protocol for a feasibility study including heart rate variability (HRV) biofeedback in people with LC was published in 2022[17] (the registration for the present study was in 2021), with the feasibility study published in 2024[18]. However, only 13 participants completed the study, limiting confidence in results. On the other hand, a commercially available platform (Visible), with a reported $1 m start-up fund and over 100,000 users is delivering biofeedback-informed energy management support. In the company's recent preprint, within-person changes in HRV and heart rate emerged as predictors of symptom change, with higher heart rate and lower HRV conferring risk for 'crashes', 'fatigue', and 'brain fog'[19]. However, key limitations, including measurement validity, sampling bias, and data accessibility render this platform unsuitable for research. However, we argue there are several limitations with this platform that render it unsuitable for research purposes. Firstly, the concepts of 'crashes', 'fatigue', and 'brain fog' are quantified without the use of validated instruments, but a 1-3 Likert

scale rating system. This means that heart rate and HRV may predict 1, 2, or 3, but whether 1, 2, or 3 is a valid representation of 'crash' (i.e. PEM) is uncertain. Secondly, as this is a consumer-focussed application, it is behind a paywall, which creates bias in 'recruitment' (they are arguably consumers rather than participants). Thirdly, HRV and heart rate data were collapsed into large time epochs, which is necessary for data analysis due to the size of the data, but this does not permit in-the-moment warnings of upcoming PEM (or 'crash'). Finally, it also suffers from ascertainment bias in that participants self-declare their impairment and as such, the dataset contains people with ME/CFS, LC, any other co-morbidities, anyone who feels tired often, and anyone who downloads the app for interest. Anybody can contribute to the Visible dataset. Therefore, a platform that warns participants of potential PEM, informed by minute-by-minute biofeedback, using validated instruments, without a subscription cost, is non-existent to date.

Therefore, the aim of this trial was to determine if activity tracking combined with just-in-time messages helped people with LC implement energy management and reduce incidents of symptom exacerbation. We hypothesised a priori that a just-in-time intervention to assist energy management (intervention group) would reduce frequency or severity of PEM compared to usual care (control).

## Results

Of 369 participants assessed for eligibility, 119 were deemed ineligible or withdrew prior to randomisation. In total, 250 participants were randomised (125 per group), but 14 controls withdrew upon group allocation, and 11 of the intervention group discontinued intervention. Thirty-four control and 30 intervention participants were lost to follow-up. Therefore, 84 intervention participants and 77 control participants completed the 6-month follow-up and post-intervention data collection for the primary endpoint and were included in final per-protocol analysis (see Supplementary Information for age and gender distributions, and for employment status at baseline). Throughout the study, 33 participants were switched to step count monitoring. In the final per-protocol analysis, this applied to 23 participants. Some participants completed the primary endpoint but not secondary outcome questionnaires, despite reminders. Therefore, the sample size for each outcome variable is reported in Tables 1 and 2. The mean and standard deviation (SD) (±) number of 100% messages received per intervention group participant in month 1, 2, 3, 4, 5, and 6 was $13 \pm 10$, $10 \pm 10$, $8 \pm 9$, $7 \pm 9$, $6 \pm 10$, and $6 \pm 10$, respectively. The mean number of days participant exceeded their 100% energy allowance over the full 6 months was $51 \pm 50$ days.

Using the sum of the DSQ-PEM questions 1–5 expressed on a 100-point scale (as per our power calculation), the intervention group value was 48 (95% CI 44–53) at baseline and 46 (95% CI 41–51) post-intervention. The control group value was 47 (95% CI 42–52) at baseline and 44 (95% CI 39–49) at follow-up (interaction effect $p = 0.614$, $\eta^2 p = 0.002$; trivial). Analysis of individual questions are reported in Table 1. There was a between group effect for question 1 frequency response, but no time or interaction effects were observed. For questions 6–8, and 10, the proportion of subjects who reported yes or no did not differ by time except for question 7 whereby the intervention group exhibited a higher proportion of yes responses pre-intervention compared to post-intervention. For question 9, there was a within-group effect for time, whereby participants felt worse after activities but there were no between group differences. There were no other observed effects.

Secondary outcomes are reported in Table 2. No interaction effects were evident, suggesting the energy management intervention was not superior to control. Some effects for time were evident, suggesting an improvement in secondary outcomes from baseline to post-intervention follow-up in both groups.

To confirm our per-protocol analysis above, we conducted intention-to-treat analysis on the primary outcome variable. For the

**Table 1 | DePaul symptom questionnaire-post-exertional malaise (DSQ-PEM) question responses for intervention and control groups, reported as per-protocol analyses using two-way mixed-model ANOVAs with condition (intervention or control) as the between-subjects factor and time (pre- and post-intervention) as a within subjects factor (questions 1–5)**

| | | Baseline mean (95% CI) | Month 6 mean (95% CI) | Repeated measures analysis of variance (ANOVA) | | |
|---|---|---|---|---|---|---|
| | | | | $p$ value time ($\eta^2 p$) | $p$ value group ($\eta^2 p$) | $p$ value interaction ($\eta^2 p$) |
| DSQ-PEM | | | | | | |
| Q1 severity | Intervention (n = 84)<br>Control (n = 77) | 2.1 (1.9–2.5)<br>2.3 (1.9–2.6) | 2.2 (1.9–2.5)<br>1.8 (1.5–2.1) | 0.072 (0.020; small) | 0.354 (0.005; trivial) | 0.058 (0.022; small) |
| Q1 frequency | Intervention (n = 84)<br>Control (n = 77) | 3.0 (2.7–3.3)<br>2.6 (2.4–2.9) | 2.8 (2.5–3.1)<br>2.4 (2.1–2.7) | 0.063 (0.022; small) | **0.039 (0.027; small)** | 0.704 (0.001; trivial) |
| Q2 severity | Intervention (n = 84)<br>Control (n = 77) | 2.1 (1.8–2.4)<br>2.2 (1.9–2.5) | 2.0 (1.7–2.3)<br>2.0 (1.7–2.4) | 0.380 (0.005; trivial) | 0.716 (0.001; trivial) | 0.656 (0.001; trivial) |
| Q2 frequency | Intervention (n = 84)<br>Control (n = 77) | 2.9 (2.6–3.2)<br>2.7 (2.4–3.0) | 2.7 (2.4–3.0)<br>2.8 (2.5–3.1) | 0.694 (0.001; trivial) | 0.711 (0.001; trivial) | 0.316 (0.006; trivial) |
| Q3 severity | Intervention (n = 84)<br>Control (n = 77) | 1.6 (1.3–2.0)<br>2.1 (1.8–2.5) | 1.9 (1.5–2.2)<br>2.0 (1.6–2.3) | 0.380 (0.005; trivial) | 0.333 (0.006; trivial) | 0.623 (0.002; trivial) |
| Q3 frequency | Intervention (n = 84)<br>Control (n = 77) | 2.5 (2.2–2.8)<br>2.5 (2.2–2.8) | 2.4 (2.2–2.7)<br>2.5 (2.1–2.7) | 0.460 (0.003; trivial) | 0.890 (0.000; trivial) | 0.856 (0.000; trivial) |
| Q4 severity | Intervention (n = 84)<br>Control (n = 77) | 2.1 (1.8–2.4)<br>2.2 (1.8–2.5) | 1.9 (1.6–2.2)<br>2.0 (1.7–2.3) | 0.311 (0.006; trivial) | 0.317 (0.002; trivial) | 0.269 (0.008; trivial) |
| Q4 frequency | Intervention (n = 84)<br>Control (n = 77) | 2.9 (2.6–3.2)<br>2.6 (2.2–2.9) | 2.7 (2.4–3.0)<br>2.7 (2.4–3.0) | 0.681 (0.001; trivial) | 0.322 (0.006; trivial) | 0.164 (0.012; small) |
| Q5 severity | Intervention (n = 84)<br>Control (n = 77) | 2.0 (1.7–2.3)<br>2.1 (1.8–2.5) | 1.9 (1.6–2.2)<br>1.8 (1.5–2.1) | 0.080 (0.019; small) | 0.861 (0.000; trivial) | 0.262 (0.008; trivial) |
| Q5 frequency | Intervention (n = 84)<br>Control (n = 77) | 2.6 (2.3–2.9)<br>2.3 (2.0–2.6) | 2.4 (2.0–2.7)<br>2.3 (2.0–2.7) | 0.552 (0.002; trivial) | 0.411 (0.004; trivial) | 0.368 (0.005; trivial). |
| | | Count (%) | Count (%) | McNemar's test<br>Within group p value ($X^2$) | | Chi squared test<br>between group p value ($X^2$) |
| Q6 response | Intervention (pre n = 84, post n = 83)<br>Control (pre n = 77, post n = 77) | 9 yes (11%)<br>75 no (89%)<br>8 yes (10%)<br>69 no (90%) | 12 yes (14%)<br>71 no (86%)<br>10 yes (13%)<br>67 no (87%) | 0.439 (0.600)<br>0.251 (1.32) | | Pre; p = 1.000 (0.0.00)<br>Post; p = 0.787 (0.073) |
| Q7 response | Intervention (pre n = 84, post n = 83)<br>Control (pre n = 77, post n = 76) | 82 yes (98%)<br>2 no (2%)<br>73 yes (95%)<br>4 no (5%) | 72 yes (87%)<br>11 no (13%)<br>68 yes (89%)<br>8 no (11%) | **0.007 (7.36)**<br>0.206 (1.60) | | Pre; p = 0.266 (1.24)<br>Post; p = 0.596 (0.280) |
| Q8 response | Intervention (pre n = 84, post n = 84)<br>Control (pre n = 77, post n = 77) | 71 yes (90%)<br>13 no (1 0%)<br>69 yes (90%)<br>8 no (10%) | 65 yes (77%)<br>19 no (23%)<br>65 yes (84%)<br>12 no (16%) | 0.157 (2.00)<br>0.206 (1.60) | | Pre; p = 0.465 (0.533)<br>Post; p = 0.258 (1.28) |
| Q 9 response | Intervention (pre n = 84, post n = 82)<br>Control (pre n = 77, post n = 76) | 1 reported ≤1 h (1%)<br>5 reported 2–3 h (6%)<br>21reported 4–10 h (25%)<br>2 reported 11–13 h (2%)<br>7 reported 14–23 h (8%)<br>48 reported ≥ 24 h (57%)<br>2 reported ≤1 h (3%)<br>5 reported 2–3 h (6%)<br>13 reported 4–10 h (17%)<br>2 reported 11–13 h (3%)<br>7 reported 14–23 h (9%)<br>48 reported ≥ 24 h (62%) | 2 reported ≤1 h (2%)<br>8 reported 2–3 h (10%)<br>15 reported 4–10 h (18%)<br>1 reported 11–13 h (1%)<br>15 reported 14–23 h (18%)<br>41 reported ≥24 h (50%)<br>2 reported ≤1 h (3%)<br>5 reported 2–3 h (7%)<br>10 reported 4–10 h (13%)<br>2 reported 11–13 h (3%)<br>10 reported 14–23 h (13%<br>47 reported ≥24 h (62%) | <0.001 (74.3)<br><0.001 (69.3) | | Pre; p = 0.861 (1.91)<br>Post; p = 0.667 (3.21) |
| Q10 response | Intervention (pre n = 82, post n = 80)<br>Control (pre n = 77, post n = 77) | 75 yes (91%)<br>7 no (9%)<br>61 yes (90%)<br>7 no (10%) | 74 yes (93%)<br>6 no (8%)<br>63 yes (86%)<br>10 no (14%) | 0.198 (4.67)<br>0.317 (1.00) | | Pre; p = 0.712 (0.136)<br>Post; p = 0.453 (1.58) |

For questions 6–10, McNemar's Test for paired samples (pre- to post-intervention) and Chi squared test for between group effects (intervention vs. control) were conducted. All analyses were two-sided. Bold font indicates $p < 0.05$.

**Table 2 | Data reported as per-protocol analyses using two-way mixed-model ANOVAs with condition (intervention or control) as the between-subjects factor and time (pre- and post-intervention) as a within subjects factor**

| | | Baseline mean (95% CI) | Post mean (95% CI) | Repeated measures analysis of variance (ANOVA) | | |
|---|---|---|---|---|---|---|
| | | | | $p$ value time ($\eta^2 p$) | $p$ value group ($\eta^2 p$) | $p$ value interaction ($\eta^2 p$) |
| **HRQoL** | | | | | | |
| SF12-physical health | Intervention (n = 53) Control (n = 50) | 54 (54–54) 54 (54–54) | 54 (54–55) 54 (54–54) | 0.754 (0.001; trivial) | 0.969 (0.000; trivial) | 0.349 (0.009; trivial) |
| SF12-mental health | Intervention (n = 53) Control (n = 50) | 59 (59–59) 59 (59–59) | 59 (59–60) 59 (59–60) | **0.003 (0.086; moderate)** | 0.241 (0.014; small) | 0.531 (0.004; trivial) |
| EQ5D anxiety | Intervention (n = 84) Control (n = 73) | 1.4 (1.2–1.6) 1.3 (1.2–1.6) | 1.2 (1.0–1.5) 1.2 (0.9–1.4) | 0.075 (0.020; small) | 0.583 (0.002; trivial) | 0.957 (0.000; trivial) |
| EQ5D mobility | Intervention (n = 84) Control (n = 73) | 1.3 (1.1–1.6) 1.5 (1.3–1.7) | 1.5 (1.2–1.7) 1.5 (1.2–1.7) | 0.416 (0.004; trivial) | 0.906 (0.000; trivial) | 0.543 (0.002; trivial) |
| EQ5D pain | Intervention (n = 84) Control (n = 73) | 1.9 (1.7–2.1) 1.8 (1.6–2.0) | 1.8 (1.6–2.0) 1.8 (1.6–2.1) | 0.721 (0.001; trivial) | 0.933 (0.000; trivial) | 0.569 (0.002; trivial) |
| EQ5D self-care | Intervention (n = 84) Control (n = 73) | 1.0 (0.8–1.2) 1.0 (0.8–1.2) | 1.0 (0.8–1.2) 0.9 (0.7–1.1) | 0.431 (0.004; trivial) | 0.803 (0.000; trivial) | 0.180 (0.012; small) |
| EQ5D usual activities | Intervention (n = 84) Control (n = 73) | 2.5 (2.3–2.7) 2.6 (2.4–2.8) | 2.3 (2.1–2.6) 2.4 (2.1–2.6) | **0.003 (0.056; small)** | 0.530 (0.003; trivial) | 0.657 (0.001; trivial) |
| EQ5D index | Intervention (n = 84) Control (n = 73) | 0.38 (0.31–0.44) 0.43 (0.37–0.49) | 0.38 (0.30–0.45) 0.39 (0.31–0.47) | 0.456 (0.004; trivial) | 0.324 (0.006; trivial) | 0.178 (0.012; small) |
| **Pain** | | | | | | |
| VAS | Intervention (n = 78) Control (n = 71) | 46 (41–51) 46 (40–51) | 46 (40–52) 44 (38–51) | 0.721 (0.001; trivial) | 0.933 (0.000; trivial) | 0.569 (0.002; trivial) |
| **Anxiety and depression** | | | | | | |
| PHQ4 | Intervention (n = 74) Control (n = 74) | 5.3 (4.7–6.0) 5.2 (4.6–5.9) | 4.7 (3.9–5.5) 4.5 (3.6–5.3) | **0.002 (0.065; small)** | 0.686 (0.001; trivial) | 0.764 (0.001; trivial) |
| **Fatigue** | | | | | | |
| FSS-7 | Intervention (n = 76) Control (n = 71) | 6.3 (6.1–6.4) 6.4 (6.3–6.6) | 6.0 (5.8–6.3) 6.0 (5.8–6.4) | **<0.001 (0.118; large)** | 0.693 (0.001; trivial) | 0.808 (0.000; trivial) |
| **Breathlessness** | | | | | | |
| MRC BQ | Intervention (n = 84) Control (n = 74) | 1.6 (1.4–1.8) 1.6 (1.4–1.7) | 1.4 (1.3–1.6) 1.5 (1.3–1.6) | **0.037 (0.028; small)** | 0.723 (0.001; trivial) | 0.369 (0.005; trivial) |
| **Self-management** | | | | | | |
| SEMCD | Intervention (n = 81) Control (n = 72) | 26 (25–28) 26 (24–28) | 31 (28–33) 31 (27–33) | 0.431 (0.004; trivial) | 0.803 (0.000; trivial) | 0.180 (0.012; small) |
| **Cognitive function** | | | | | | |
| SDMT TC (/60) | Intervention (n = 82) Control (n = 77) | 58 (57–59) 58 (57–59) | 59 (59–60) 59 (58–59) | 0.154 (0.013; small) | 0.355 (0.005; trivial) | 0.107 (0.016; small) |
| SDMT TTCA (s) | Intervention (n = 82) Control (n = 77) | 133 (126–141) 146 (129–163) | 132 (113–151) 127 (120–134) | 0.250 (0.008; trivial) | 0.476 (0.003; trivial) | 0.119 (0.015; small) |
| SDMT TPCA (s) | Intervention (n = 82) Control (n = 77) | 2.3 (2.2–0.5) 2.5 (2.2–2.9) | 2.2 (1.9–2.6) 2.2 (2.0–2.3) | 0.154 (0.013; small) | 0.355 (0.005; trivial) | 0.107 (0.016; small) |

All analyses were two-sided. Bold font indicates $p < 0.05$.
*Sf12* 12-item short form health survey, *EQ5D* EuroQol 5-dimension health questionnaire, *Pain VAS* pain visual analogue scale, *PHQ4* patient health questionnaire-4, *FSS* (7 item) fatigue severity scale (7-item version), *MRC BQ* medical research council breathlessness questionnaire, *SEMCD* self-efficacy to manage chronic disease, *SDMT* symbol digit modalities test, *TC* total correct (out of 60), *TTCA* total time for correct answers only, *TPCA* average time per correct answer.

DSQ-PEM sum score (visualised in Fig. 1), the mixed-effects model revealed no effect of time ($p = 0.097$), group ($p = 0.195$), or interaction effect ($p = 0.621$). Questions 1–5 responses are visualised in Fig. 2. For DSQ-PEM Q1 severity, the mixed-effects model revealed no effect of time ($p = 0.414$), group ($p = 0.926$), or interaction effect ($p = 0.053$). For DSQ-PEM Q1 frequency, the mixed-effects model revealed no effect of time ($p = 0.052$), group ($p = 0.813$), or interaction effect ($p = 0.423$). For DSQ-PEM Q2 severity, the mixed-effects model revealed no effect of time ($p = 0.454$), group ($p = 0.147$), or interaction effect ($p = 0.792$). For DSQ-PEM Q2 frequency, the mixed-effects model revealed no effect of time ($p = 0.154$), group ($p = 0.099$), or interaction effect ($p = 0.639$). For DSQ-PEM Q3 severity, the mixed-effects model revealed no effect of time ($p = 0.309$), or interaction effect ($p = 0.489$), but there was an effect of group ($p = 0.016$), favouring intervention. Tukey's post hoc test revealed a difference between groups at month 2 only ($p = 0.015$), favouring intervention. For DSQ-PEM Q3 frequency, the mixed-effects

model revealed no effect of time ($p = 0.107$), or interaction effect ($p = 0.584$), but there was an effect of group ($p = 0.038$), favouring intervention. Tukey's post hoc test revealed no differences between groups at any specific month. For DSQ-PEM Q4 severity, the mixed-effects model revealed no effect of time ($p = 0.496$), group ($p = 0.580$), or interaction effect ($p = 0.309$). For DSQ-PEM Q4 frequency, the mixed-effects model revealed no effect of time ($p = 0.391$), group ($p = 0.650$), or interaction effect ($p = 0.961$). For DSQ-PEM Q5 severity, the mixed-effects model revealed no effect of time ($p = 0.174$), group ($p = 0.182$), or interaction effect ($p = 0.739$). For DSQ-PEM Q5 frequency, the mixed-effects model revealed no effect of time ($p = 0.591$), group ($p = 0.455$), or interaction effect ($p = 0.618$).

To confirm the tests of mean differences above (i.e. ANOVA) were robust, we determine the number of individuals considered positive for PEM (i.e. discrete yes/no). A score of ≥2 for both frequency and severity on a specific item (Q1–5) suggests PEM is both present and

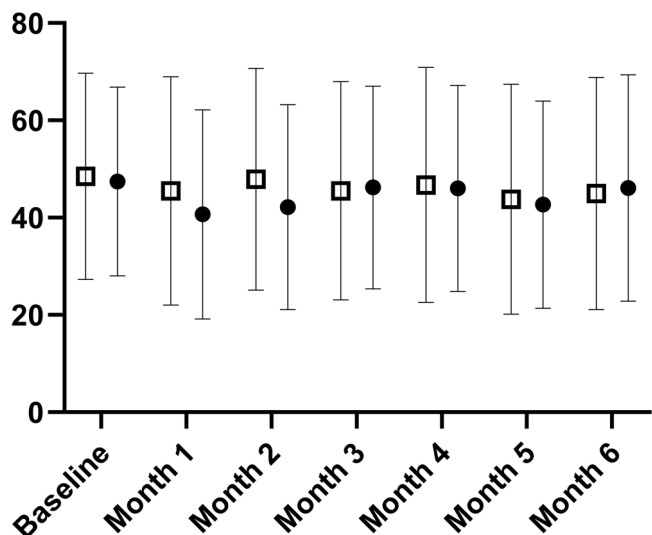

**Fig. 1 | DePaul symptom questionnaire-post-exertional malaise (DSQ-PEM) sum score for intervention (closed circles) and control (open squares) groups, following as intention-to-treat analyses at each month.** Data are presented as mean ± standard deviation and units are arbitrary. $N = 231$. The linear mixed-effects model was used to test for differences. All analyses were two-sided.

clinically relevant[4]. Therefore, participants could be either positive or negative for PEM. When groups were combined, McNemar's Test revealed the number of individuals classified as PEM positive decreased from baseline to follow-up ($p = 0.004$). 111 participants were considered PEM positive at baseline and follow-up, 31 participants were PEM positive at baseline but not at follow-up (improvement), 12 participants were PEM negative before but positive after (worsening), and seven participants were PEM negative at both time points. In the intervention group, McNemar's Test revealed no change in PEM status from baseline to post-intervention ($p = 0.394$). Fifty-nine participants were considered PEM positive at baseline and post-intervention, 13 participants were PEM positive at baseline but not at post-intervention (improvement), nine participants were PEM negative before but positive after (worsening), and three participants were PEM negative at both time points. In the control group, the number of individuals classified as PEM positive decreased from baseline to follow-up ($p = 0.001$). Fifty-two participants were considered PEM positive at baseline and follow-up, 18 participants were PEM positive at baseline but not at follow-up (improvement), three participants were PEM negative before but positive after (worsening), and four participants were PEM negative at both time points. We subsequently used Chi squared test for between group effects, which exhibited no differences at baseline ($p = 0.411$) or follow-up ($p = 0.155$).

To confirm our results were relevant for both genders, we disaggregated the results. In females only, repeated measures ANOVA

**Fig. 2 | DePaul symptom questionnaire-post-exertional malaise (DSQ-PEM) question responses for intervention (closed circles) and control (open squares) groups, following as intention-to-treat analyses at each month.** *Indicates pairwise difference at the $p = 0.016$ level. Data are presented as mean ± standard deviation and units are arbitrary. $N = 231$. The linear mixed-effects model was used to test for differences, and post-hoc location of differences was investigated using the Tukey post-hoc test and all analyses were two-sided.

revealed an effect of time ($p = 0.021$, $\eta^2 p = 0.041$; small), but not group ($p = 0.825$, $\eta^2 p = 0.000$; trivial), or interaction effect ($p = 0.358$, $\eta^2 p = 0.007$; trivial) for the primary outcome variable (sum of the DSQ-PEM). The intervention group value was 50 (95% CI 45–55) at baseline and 43 (95% CI 38–49) post-intervention. The control group value was 49 (95% CI 43–54) at baseline and 46 (95% CI 40–52) at follow-up. In males only, repeated measures ANOVA revealed no effect of time ($p = 0.275$, $\eta^2 p = 0.038$; small), or group ($p = 0.057$, $\eta^2 p = 0.112$; moderate), but there was an interaction effect ($p = 0.045$, $\eta^2 p = 0.123$; moderate) for the primary outcome variable. The intervention group value was 43 (95% CI 33–53) at baseline and 56 (95% CI 46–65) post-intervention. The control group value was 41 (95% CI 30–52) at baseline and 37 (95% CI 27–46) at follow-up.

As suggested by a reviewer, we also analysed results without those who were switched to steps rather than heart rate. Repeated measures ANOVA revealed no effect of time ($p = 0.213$, $\eta^2 p = 0.011$; small), group ($p = 0.998$, $\eta^2 p = 0.000$; trivial), or interaction effect ($p = 0.776$, $\eta^2 p = 0.001$; trivial) for the primary outcome variable.

## Discussion

The aim of this trial was to determine if activity tracking combined with just-in-time messages helped people with LC implement energy management to reduce PEM frequency or severity. We hypothesised a priori that a just-in-time intervention to assist energy management (intervention group) would reduce frequency or severity of PEM compared to usual care only (control). The main finding of the present study was that a just-in-time intervention in conjunction with a wearable activity tracker to support energy management in people with LC was not more effective than usual care. There were no intervention effects on DSQ-PEM sum score, or for questions 1–5. Both groups experienced an improvement in secondary outcomes, including mental health, severity of problems in usual activities, anxiety and depression, fatigue, and breathlessness at follow up. This indicates regression to the mean, whereby both groups improved over time. This is further supported by our sensitivity analysis confirming a change in PEM positive cases from baseline to post-intervention. As a result, we must reject our hypothesis that a just-in-time intervention to assist energy management would reduce frequency or severity of PEM compared to controls. Nevertheless, the intervention group reported fewer 'yes' responses to question 7 'do you experience a worsening of your fatigue/energy related illness after engaging in minimal physical effort?' over time, while no differences were observed for the control group. Additionally, responses to question 9 whereby participants indicated whether they felt worse after activity indicated an improvement over time in the intervention group. This suggests that the intervention group perceived less PEM symptoms due to physical effort over the course of the intervention. Moreover, the intention to treat sensitivity analysis suggested the intervention group had preferential responses to question 3 ('mentally tired after the slightest effort') compared to controls. However, given these findings were not corroborated by other items from the DSQ-PEM these findings should be interpreted with caution.

This is the first study of its kind to examine technology to support energy management in conditions with PEM. We rigorously tested energy management in this randomised controlled trial (RCT), the first since the landmark PACE Trial, published over a decade ago[20]. There are several factors that may explain the lack of intervention efficacy. At the time of study development, LC was a relatively new condition, with most individuals having been initially infected during the first wave of COVID-19. There was an emerging view that LC symptoms were similar to other post-viral conditions such as ME/CFS, such that they were almost indistinguishable as separate conditions[21]. Consequently, the intervention was developed under the assumption that LC would resemble ME/CFS and that individuals would experience prolonged and potentially persistent symptoms. This assumption holds for a subset of LC patients. For instance, members of our PPI group have been managing PEM and other symptom exacerbations since their initial infection in 2020. However, new evidence suggests that most individuals recover from LC, albeit very slowly, over a period of several months. For example, we previously tracked 287 individuals for 9 months after their initial COVID-19 infection, generating early prospective data on the transition from acute infection to LC[22]. In that study, most participants experienced symptoms for four to seven months, with around 3% experiencing longer-lasting symptoms at follow-up. Similarly, Oliveira et al.[23] tracked 34 LC patients for 12 months, finding that many initially met the case definition for ME/CFS, but most showed symptom improvement by follow-up. Jason et al.[24] also found that over five months, most LC patients reported improvements in symptoms, particularly sleep and incidence of PEM. These data are supported by the present investigation whereby we noted a reduction in PEM positive cases in the whole cohort from baseline to follow-up. A consequence of these findings is that, for most individuals (excluding a small proportion who may develop a more persistent post-viral condition), LC differs from conditions like ME/CFS, where recovery can take years or may not occur at all. Consequently, the effectiveness of this type of activity-tracking just-in-time intervention is less certain for individuals with ME/CFS, who are unlikely to experience significant recovery or symptom reduction within the timeframe of such trials.

There are some limitations of this trial that should be noted. At the time of recruitment, there was no agreed case definition for LC. Moreover, several of our participants had LC from presumed COVID-19 infection early in the pandemic, before home testing kits were available. We excluded hospitalised individuals who may have undergone PCR confirmation early in the pandemic. Consequently, definitive diagnosis of the participants' condition was not possible. While ascertainment bias is both common and well-established in LC research[25], it remains a limitation that should be considered. Nevertheless, robust randomisation and allocation concealment should prevent systematic effect in the resulting data. Secondly, participants' mental exertion was not considered in the energy management strategy. Managing mental exertion is of course a central aspect of energy management. In future research, consideration should be given to the inclusion of mental exertion in energy management assessments, although this could increase participant burden so a passive measure of mental exertion would be beneficial. Thirdly, we did not assess participants' psychological or affective states (e.g. motivation, mood, stress), which may influence physical activity behaviour, energy management decisions, and susceptibility to PEM. A fourth limitation, although necessary, was that we provided activity trackers to the intervention group but not the control group. The reason we believed this necessary was provision of a wearable would be a form of intervention in itself. As such, it is possible that similar frequency and severity of PEM existed between the two groups despite differences in physical activity, but due to our design we cannot examine this. Future work may wish to attempt to control for this limitation. On the topic of wearable devices, Fitbits have demonstrated reasonable validity for step counting and heart rate tracking in healthy adults, but their accuracy diminishes at slower walking speeds[26] and higher intensity activity[27]. Therefore, some episodes of overexertion may not have been adequately detected. Moreover, it is possible that some control participants used their own activity tracker or app to modify and manage their activity. For example, part way through the trial a commercial app aimed at energy management was launched and some control participants may have decided to use that app, introducing contamination bias. Fifth, while two-arm RCTs are appropriate for determining efficacy of an intervention package, they are less helpful in providing data for optimisation, such as identifying which specific components of an intervention are influencing the outcomes of interest. This optimisation is crucial for enhancing efficiency and

effectiveness[28], which is often only achieved through repeated testing and refinement, or a factorial trial. Our qualitative work indicated participants viewed the Pace Me platform positively, gaining a sense of insight, validation, and control over their condition[29]. Therefore, although our framework was not successful at reducing mean PEM values in people with LC in 2022 and 2023, it should be refined and further developed for deployment in different settings or populations, much like the model of drug repurposing. Moreover, as behaviour change interventions alone often produce small, inconsistent effects[30], individual-level analysis may reveal a subset of individuals who benefitted from this intervention.

Despite the limitations described above, the present study had a significant number of strengths. Firstly, we tested a novel method of energy management utilising technology, meaning this study is transformative compared with previous work, which relied on paper-based journalling and activity planning. This project provided a user-friendly platform that simplified complex tasks of tracking activity, comparing PEM events, and influencing daily activity decisions for the end user. We hope that despite the primarily null findings herein, this energy management framework can be employed for other PEM-experiencing conditions such as lupus, ME/CFS, multiple sclerosis, and rheumatoid arthritis. Advances in digital technologies have opened unprecedented opportunities to deliver effective and scalable behaviour change interventions and just-in-time adaptive interventions have existed for almost a decade[31,32], but until now, had not been applied to conditions with PEM. Therefore, implementing just-in-time energy management frameworks in these conditions could have significant impact given the scalability and inclusivity of remote support[33]. This step-wise change in how energy management can be delivered could result in positive individual, societal, and economic impacts.

As mentioned in the introduction, a commercially available platform (Visible), is delivering biofeedback-informed energy management support. In the company's recent preprint, within-person changes in HRV and heart rate emerged as predictors of symptom change, with higher heart rate and lower HRV conferring risk for 'crashes', 'fatigue', and 'brain fog'[19]. However, key limitations, including measurement validity, sampling bias, and data accessibility render this platform unsuitable for research. However, we argue there are several limitations with this platform that render it unsuitable for research purposes. Firstly, the concepts of 'crashes', 'fatigue', and 'brain fog' are quantified without the use of validated instruments, but a 1–3 Likert scale rating system. This means that heart rate and HRV may predict 1, 2, or 3, but whether 1, 2, or 3 is a valid representation of 'crash' (i.e. PEM) is uncertain. Secondly, as this is a consumer-focussed application, it is behind a paywall, which creates bias in 'recruitment' (they are arguably consumers rather than participants). Thirdly, HRV and heart rate data were collapsed into large time epochs, which is necessary for data analysis due to the size of the data, but this does not permit in-the-moment warnings of upcoming PEM (or 'crash'). Finally, it also suffers from ascertainment bias in that participants self-declare their impairment and as such, the dataset contains people with ME/CFS, LC, any other co-morbidities, anyone who feels tired often, and anyone who downloads the app for interest. Anybody can contribute to the Visible dataset. Therefore, we argue that a platform that warns participants of potential PEM, informed by minute-by-minute biofeedback, using validated instruments, without a subscription cost, is non-existent to date.

In conclusion, this study found a lack of treatment effect for those who received a just-in-time energy management intervention. Importantly, there were no adverse incidents in the intervention group, addressing a potential concern raised by our PPI group regarding the use of activity trackers in this population. This equivalence, along with the absence of adverse incidents, demonstrates that the intervention was safe and feasible. Future work should explore efficacy of similar apps in patients with conditions that are not expected to recover, such as ME/CFS and should explore the effects of gender on results observed.

## Methods

### Study design

The study was designed as a pragmatic single-centre RCT. The intervention was a remotely delivered, just-in-time energy management support programme designed to reduce the frequency or severity of PEM experienced by individuals experiencing LC. The trial ran between January 2022 and September 2023; recruitment ran between November 2021 and February 2023. The study was approved by the University of the West of Scotland Institutional Ethics Board (approval number 16638) and the trial was registered with ICTRN (ISRCTN16033549; https://doi.org/10.1186/ISRCTN16033549). Informed consent was obtained from all participants.

### Participants

**Inclusion criteria.** Eligible participants had to be adults (≥18 years) reporting persistent symptoms following a COVID-19 infection which interfered with daily activities (in line with NICE guidelines 2021) and experiencing PEM. Many of those eligible to participate in the trial had contracted COVID-19 before home-based testing kits were available and before polymerase chain reaction (PCR) testing was made publicly available. Consequently, we accepted self-reported persistent symptoms of ≥8 weeks following an infection consistent with COVID-19. Participants had to recover at home (e.g. access to their GP but no ongoing clinical care related to their COVID-19 infection) and have access to a smartphone with Android version 6 or iOS version 10 or higher.

**Exclusion criteria.** Participants were excluded if they (1) had insufficient English language to understand messages, (2) had no smartphone access, (3) were participating in another LC intervention, (4) had impaired cognitive function which compromises comprehension of study information or messaging, (5) were receiving therapies known to cause symptom exacerbations (e.g. chemotherapy) or aimed at treating LC, (6) could not wear a FitBit at home or work, (7) had another fatigue-related condition (e.g. CFS), or (8) were receiving ongoing care for LC via primary or secondary care services.

**Recruitment.** Individuals expressed interest to trial information on social media by sharing their email address, or contacting our PPI partner (LC Scotland), or contacting us directly via email. We emailed interested parties an information sheet and asked them to review the material and respond after 48 h. If there was no response, we followed up with a reminder email asking if they were still interested in participating. Those who confirmed their interest were scheduled for an online meeting or a phone call, during which we reviewed inclusion and exclusion criteria, provided a brief overview of the study, and offered time for questions about the trial. Participants chose whether they wished to enrol in the study, and we obtained their verbal consent to proceed with randomisation during that meeting. Recruitment was assisted by our partner organisation, LC Scotland, and involved promotion of the trial via social media groups, print media, a study website, and meetings with LC Scotland members. The trial targeted people who had not been hospitalised following their COVID-19 infection, therefore there was no data linkage or recruitment via primary or secondary care.

### Randomisation and masking

A secure third-party service (studyrandomizer.com) was used to randomise 250 participants 1:1 into two evenly distributed study arms (intervention or control) with each arm consisting of 125 participants (50% of the total sample; equal allocation). Randomisation into the respective arms was conducted at the first meeting (Fig. 3) after

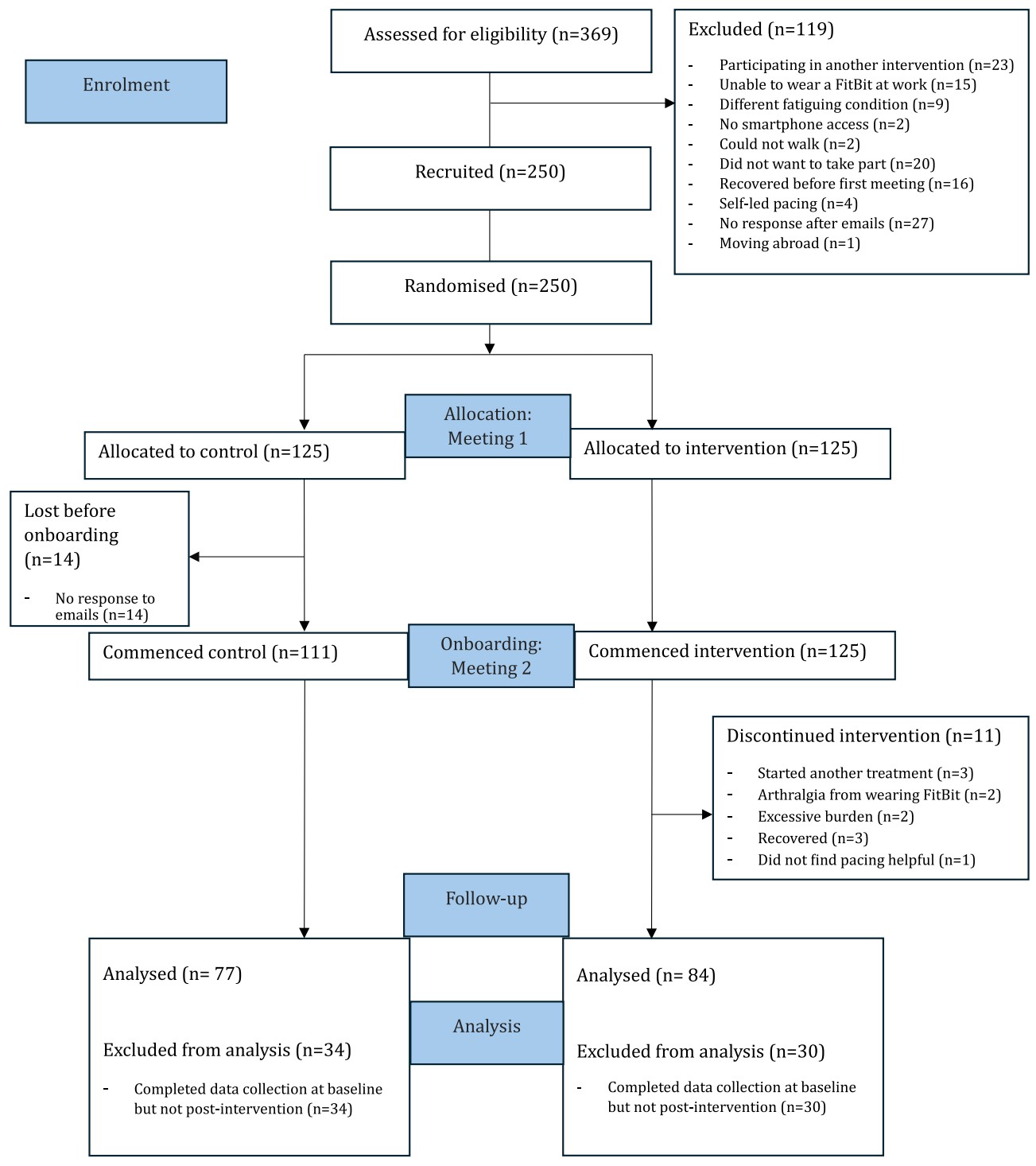

**Fig. 3 | CONSORT flow diagram of study recruitment.**

enrolment using a permuted block algorithm, with a fixed block size. Participants randomised to the intervention group received usual care with just-in-time messaging support, and the control group received usual care only. NS-H was the senior post-doctoral trial manager and generated the sequence, enroled participants, and assigned them to the trial groups. Participant blinding was impossible due to the requirements of the intervention group.

## Procedures
Following randomisation at meeting 1, participants were contacted via email after the meeting to arrange meeting 2 for onboarding,

approximately 1 week later depending on participant availability. This provided time for an activity tracker (Fitbit Charge 5, Fitbit Inc., San Francisco, CA, USA) to be posted to intervention group participants. At the second meeting, all participants downloaded the study app ('Pace Me'), while only the intervention group also downloaded the FitBit app. The intervention group's version of the study app included energy management support and data collection instruments, while the control group's version included only the data collection instruments. NS-H guided participants through the app onboarding process, which involved creating an account on the study app, logged in, and provided additional electronic consent to ensure they understood their

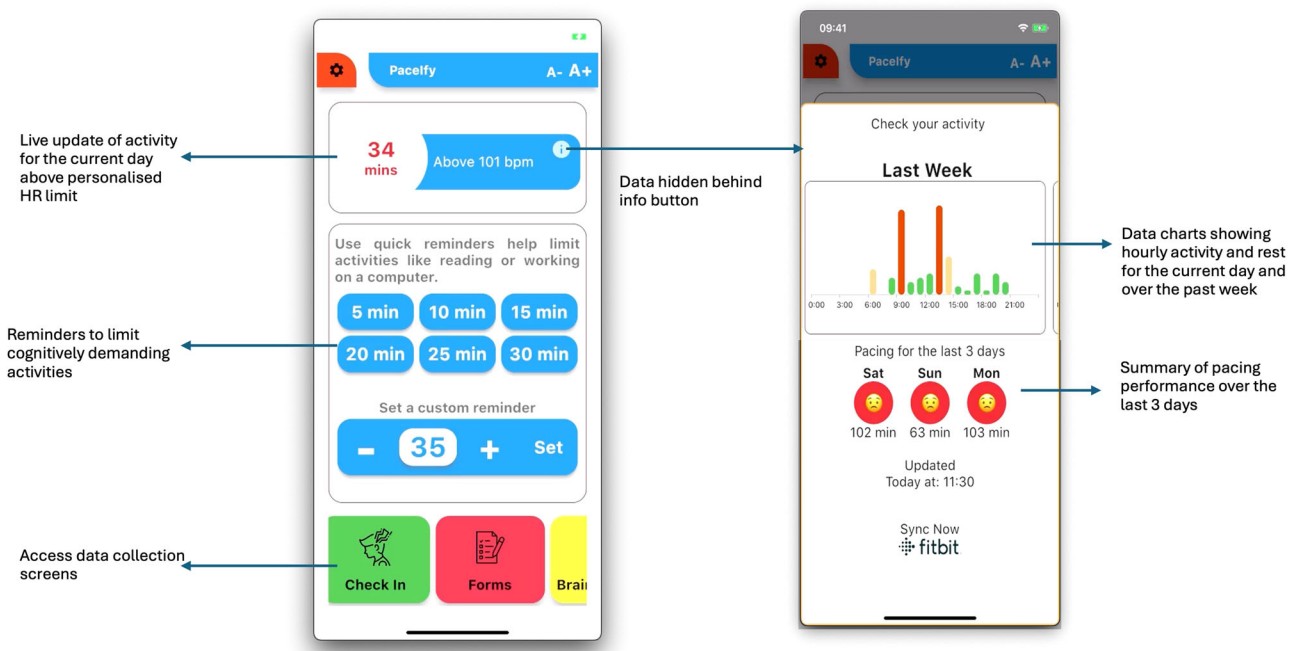

**Fig. 4 | Example of the in-app display for the intervention group with energy management support.**

interaction with the app. Intervention participants were assisted to disable FitBit app notifications. Screenshots of completed checkboxes and digital signatures were captured and securely stored separately from other study data. Gender was self-reported at the initial enrolment meeting and was not a factor in the study design. Intervention participants entered their corresponding activity tracker number to activate the intervention features of the app (including real time tracking of energy management), while control participants only had access to in app questionnaires (Fig. 4). At meeting 2, participants were encouraged to fill in the questionnaires as soon after the meeting as possible. We checked the database one week later and contacted them if they had not completed the questionnaires to ensure baseline data capture before starting the intervention or control part of the experiment. All participants received automated alerts to remind them when they were due to complete specific questionnaires and reminders if they remained uncompleted a week after the intended completion time. Participants were contacted at the end of the study to complete all questionnaires before returning their Fitbit or being provided with the energy management platform as a thank you for participation.

### Intervention design

The Pace Me app aimed to support energy management in individuals with LC. The intervention design was informed by the Behaviour Change Wheel[35], evidence from literature[11,36], and PPI feedback. The final design used goal setting, action planning, self-monitoring, feedback, prompts, and personalisation to assist users in managing energy. The intervention integrated heart rate (and step count if needed) feedback using Fitbit Charge 5, logging of PEM, logging of symptoms, and support messaging when patients exceeded their energy allowance (see Supplementary Information or full details of the intervention implementation, and for a data pipeline visualisation).

The intervention used a personalised activity allowance, initially set to no more than 30 min of activity above 60% of their age-predicted heart rate maximum. This allowance was iteratively adjusted in response to participants' activity and PEM reports. If participants reported PEM and had exceeded the allowance, the allowance remained unchanged (i.e. the PEM may have been due to poor energy

management). If participants reported PEM but had remained within their allowance for the preceding 3 days, the allowance was reduced in a stepwise fashion (i.e. activity allowance may be too large to prevent PEM). Conversely, the allowance was increased if participants did not report PEM for three consecutive weeks. For participants who requested to be changed from heart rate, step counts were used to set activity allowance. These limits were not intended as targets but as a guide for energy management. During the demonstration of the app and pacing feature, participants were informed that if they did not receive a notification indicating they had reached 50% of their daily activity allowance, they were on track. Pacing notifications were only sent to prompt users to 'slow down'. This understanding was evident in the emails we received, in which some participants apologised for overexerting themselves when they received a 100% notification. Within the app, there was a 'request a call back' function, and it was explained this was how participants could report adverse effects.

The Pace Me app featured a home screen that displayed real-time tracking showing how much of that day's energy allowance participants had used. Visualisations and emojis showed when participants were most at risk of exceeding their limits and provided feedback to help participants understand their pacing patterns. Participants received app notifications as their activity exceeded specific thresholds, capped at three notifications per day to avoid overwhelming users. Notifications were triggered when participants reached 50%, 75%, and 100% of their activity allowance. They included an alert message and support tips sourced from individuals with experience in pacing strategies for managing energy (Fig. 5). These images were created using the free version of Canva (Browser version; Sydney, Australia).

### Outcomes

The primary outcome was frequency and severity of PEM assessed using the DePaul Symptom Questionnaire-PEM (DSQ-PEM)[4], pre- and post-intervention. Questions 1–5 were measured on a five-point Likert scale with a 'frequency' domain (0 = none of the time, 1 = a little of the time, 2 = about half the time, 3 = most of the time, and 4 = all of the time) and a 'severity' domain. (0 = symptom not present, 1 = mild, 2 =

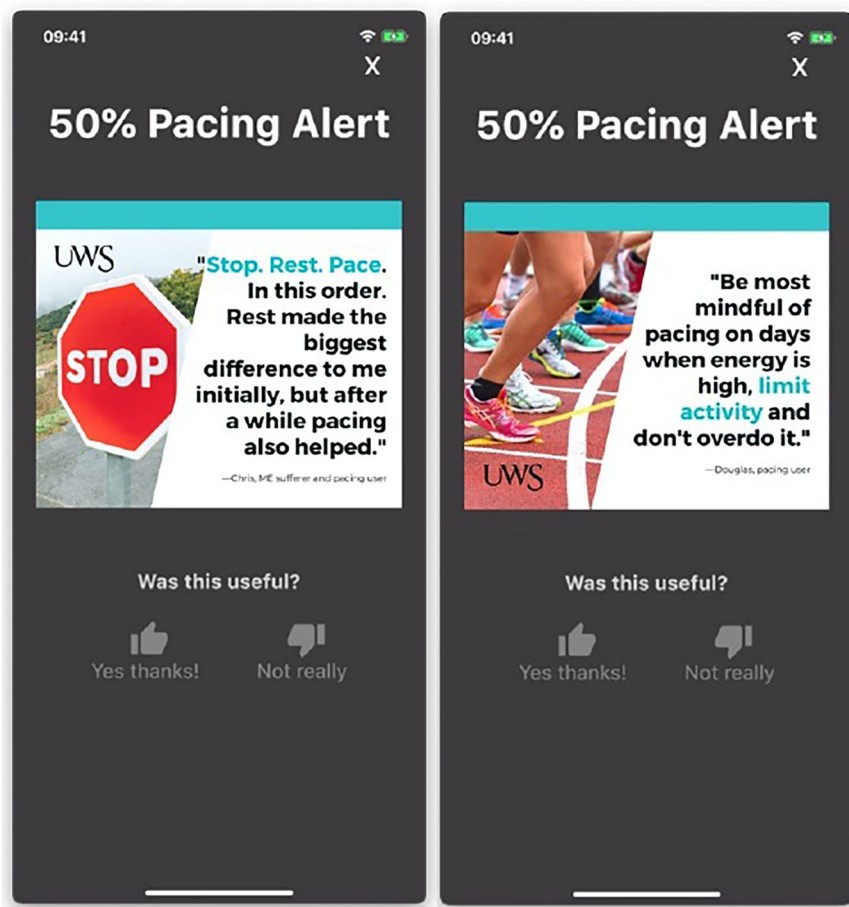

**Fig. 5 | Two examples of energy management alerts received by the intervention group.** The image displays the alerts when expanded (i.e. pressed on the phone home screen and opened the Pace Me app).

moderate, 3 = severe, and 4 = very severe). Questions 6–8 and 10 were dichotomous yes/no responses, and question 9 asked 'if you feel worse after activities, how long does this last?' with six options: ≤1 h, 2–3 h, 4–10 h, 11–13 h, 14–23 h, or ≥24 h.

Secondary outcomes were the 12-Item Short Form Health Survey, the EuroQol 5-Dimension Health Questionnaire, Pain Visual Analogue Scale, Patient Health Questionnaire-4, Fatigue Severity Scale (7-item version), Medical Research Council Breathlessness Questionnaire, Symbol Digit Modalities Test total correct (out of 60), total time for correct answers only, and average time per correct answer.

## Statistical analysis

To determine sample size, our primary outcome variable was the DSQ-PEM. Using previous work, a minimum clinically relevant difference can be estimated as a change of 13 points on a 100-point scale[37]. Assuming a SD of 25[37], this resulted in an effect size of $f = 0.25$. We calculated our desired sample size for a two-way mixed-model (within- and between-subjects) analysis of variance (ANOVA). Using the Web-Power package in R Studio (version 2024.04.2 + 764), and the wp.rmanova function, with two groups, two time points, $f = 0.25$, assuming sphericity, $\alpha = 0.05$, $1 - \beta = 0.9$, testing for an interaction effect, the total $n$ was 170 (85 per group). Consequently, we aimed to recruit 125 participants per group to allow for 30% drop-out.

All analyses were conducted using Jamovi version 2.3.21. Data were tested for normal distribution and homogeneity of variance to confirm parametric assumptions were met. Data are presented in text and tables as means and 95% confidence intervals (CI) unless otherwise stated. Because of randomisation, we did not undertake analysis of

baseline equivalence, since the null hypothesis must be true and any differences due to chance[38]. Only participants who completed follow-up testing were included in the primary analysis (i.e. per protocol analysis). The effect of the intervention on primary and secondary outcomes was examined using two-way mixed-model ANOVAs with condition (intervention or control) as the between-subjects factor and time (pre- and post-intervention) as a within subjects factor. To confirm our assumptions were robust, we conducted sensitivity analysis in the form of intention to treat analysis for all participants who were randomised. In the case of missing data, mixed-effects models were used to account for the repeated measures structure of the data and to provide robust handling of missing data. Unlike non-parametric methods, which require complete case analysis and often assume homogeneity of variance across groups, mixed-effects models offer greater flexibility by allowing the inclusion of data under the assumption that data are missing at random. The linear mixed-effects model was specified for DSQ-PEM sum score (primary outcome variable), and DSQ-PEM questions 1–5, with time (seven monthly time points) and group (intervention and control) as fixed effects and subject ID included as a random intercept to account for individual variability. No outcome distributions deviated from normality, so no data transformations were applied. In the case of main effects, location of differences was further investigated using the Tukey post-hoc test. Alpha level is reported as exact $p$ values and not described dichotomously as 'significant' or otherwise as recommended by the American Statistical Association[39]. We expressed effect sizes from the ANOVA as partial eta-squared ($\eta^2 p$), with values of 0.01, 0.06, and 0.14 interpreted as small, moderate, and large, respectively[40]. For categorical

data, (DSQ-PEM questions 6-10, and DSQ-PEM screening as positive or negative) we used McNemar's Test for paired samples (pre- to post-intervention), or Chi squared test for between group effects (intervention vs. control).

### Reporting summary

Further information on research design is available in the Nature Portfolio Reporting Summary linked to this article.

## Data availability

Data collected for this study, including individual anonymised participant data, is available via figshare: https://doi.org/10.6084/m9.figshare.30053614. The primary outcome data generated in this study have been deposited in the ISCTN database here: https://doi.org/10.1186/ISRCTN16033549[34].

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

## Acknowledgements

This study was funded by the National Institute for Health and Care Research (NIHR) (Research Award COV-LT2-0010). The views expressed in this publication are those of the authors and not necessarily those of NIHR or the Department of Health and Social Care.

## Author contributions

Conceptualisation: Nicholas F Sculthorpe; Methodology: Nilihan EM Sanal-Hayes, Lawrence D Hayes, Jacqueline L Mair, Joanne Ingram, Jane Ormerod, David Carless, Natalie Hilliard, Marie Mclaughlin, Rachel Meach, Nicholas F Sculthorpe; Formal analysis and investigation: Nilihan EM Sanal-Hayes, Nicholas F Sculthorpe; Investigation: Nilihan EM Sanal-Hayes, Nicholas F Sculthorpe; Resources: Nilihan EM Sanal-Hayes, Lawrence D Hayes, Jacqueline L Mair, Joanne Ingram, Jane Ormerod, David Carless, Natalie Hilliard, Marie Mclaughlin, Rachel Meach, Nicholas F Sculthorpe; Writing-original draft preparation: Nilihan EM Sanal-Hayes, Lawrence D Hayes, Jacqueline L Mair, Antonio Dello Iacono, Joanne Ingram, Nicholas F Sculthorpe; Writing-review and editing: Nilihan EM Sanal-Hayes, Jacqueline L Mair, Lawrence D Hayes, Antonio Dello Iacono, Joanne Ingram, Jane Ormerod, David Carless, Natalie Hilliard, Marie Mclaughlin, Rachel Meach, Nicholas F Sculthorpe; Visualisation: Nicholas F Sculthorpe; Supervision: Nicholas F Sculthorpe; Project administration: Nilihan EM Sanal-Hayes, Lawrence D Hayes, Jacqueline L Mair, Antonio Dello Iacono, Joanne Ingram, Jane Ormerod, David Carless, Natalie Hilliard, Marie Mclaughlin, Rachel Meach, Nicholas F Sculthorpe; Funding acquisition: Lawrence D Hayes, Jacqueline L Mair, Joanne Ingram, Jane Ormerod, David Carless, Natalie Hilliard, Nicholas F Sculthorpe.

## Competing interests

The authors declare no competing interests.
