## [Peer Review File · Nature Communications]

A Digital Platform with Activity Tracking for Energy Management Support in Long COVID: A Randomised Controlled Trial

Corresponding Author: Dr Lawrence Hayes

Version 0:

Reviewer comments:

Reviewer #1

(Remarks to the Author)

This is a thoughtful well-written effort to test a mobile device for energy self-management in a pragmatic trial in people with Long COVID. Intervention guidance is based on self-report PEM in relation to objective heart rate and step counts. This guidance yielded Just-in-time messages to increase subject awareness of opportunities to adjust activity levels to reduce and better manage self-report PEM.

The authors did not seek to determine at post-intervention if messages were actually perceived by subjects as “just in time,” an important element of usability. Given no significant improvement in the active intervention group despite fairly granular management feedback to subjects, post-intervention feedback from participants might have helped to identify potentially correctable issues with the protocol.

Also, data about psychological and affective states (motivation, mood, stress levels) which can influence physical activity, sedentary behavior and PEM were not collected. For instance, patients with ME/CFS and Long COVID may disregard symptom-based signals (e.g., increased headache) of upcoming crashes, a manifestation of severe PEM. Ignoring such signals may be driven by self-defeating motivation to stay as active as possible in the moment and wishful thinking about avoiding a subsequent crash. Identifying these affective states during an intervention may increase patient awareness of how they can and do undermine energy self-management.

No adherence data were presented. Although “guidance” rather than direct instructions to modify activity levels was provided, it is unclear how often the guidance offered was actually followed. For instance, was a guidance message that activity level was too high followed by a reduction in steps? If such guidance was ignored, this also suggests the importance of collaborative sessions with study subjects regarding study design.

Although Energy Management is the central construct and intervention goal, it is not clear how energy was measured. No definition was given. The authors noted that Energy Management requires participants to recall and compare two poorly described constructs; ‘energy availability’ and ‘energy use’, especially problematic when dealing with impaired patients. It seems that energy was used as more of an appealing intervention label rather than an operationalized construct. This issue should be addressed.

There may be no published peer review data on the use of digital technology for energy management in CFS, but the commercial company Visible has collected apparently enormous amounts of mobile HR/HRV and activity monitor data in Long COVID and CFS patients in order to generate pacing suggestions via their wearable app. It might inform the authors’ thinking about their mobile app trial to communicate with this company. A recently posted paper apparently based on Visible technology data on Research Square (Aitken et al; DOI: 10.21203/rs.3.rs-5423422/v1) provides predictive models of HR and HRV in relation to symptom exacerbations in ME/CFS and Long COVID using data from mobile device technology.

Detail

Line 65; myalgic encephalitis—misspelled. Should be: Encephalomyelitis

Line 103. The trial ran between January 2022 and September 2023. But the paper later states: At the time of recruitment, the UK was engaged in COVID-19 related lockdowns. Please clarify. Lockdowns ended in 2020.

What are the noteworthy results?

This is an initial controlled trial of mobile digital technology consisting of age-predicted heart rate maximum, step counts and PEM reporting to help guide and manage activity levels that maximize physical function and moderate risk of triggering PEM.

- Will the work be of significance to the field and related fields? How does it compare to the established literature? If the work is not original, please provide relevant references.

The trial does break new ground in the sense that no previously published trials have used this algorithmic approach in Long COVID or ME/CFS. The intervention combines objective measures and symptom self-report to improve illness management all housed in a wearable device. Although there is no published scientific literature on this, a popular commercial device, i.e., Visible, takes a very similar approach to Long COVID and ME/CFS self-management. A reference to this product would indicate a more thorough review of such devices.

- Does the work support the conclusions and claims, or is additional evidence needed?

The data does support its conclusions which are appropriately modest given null findings.

- Are there any flaws in the data analysis, interpretation and conclusions? - Do these prohibit publication or require revision? Revision is required to more thoroughly consider the methodological limitations of the study as indicated above.

- Is the methodology sound? Does the work meet the expected standards in your field?

The methodology is consistent with the standards of a pragmatic RCT.

- Is there enough detail provided in the methods for the work to be reproduced?

Yes. The methodology contains sufficient detail to be reproduced. Given the technicals of a device-based intervention some consultation would probably be necessary.

Reviewer #2

(Remarks to the Author)

General comments

As a reviewer, I had the privilege of reviewing the paper "A Digital Platform with Activity Tracking for Energy Management Support in Long COVID: A Randomized Controlled Trial".

The present study examines the effects of a 6-month just-in-time intervention to support energy management in Long COVID (LC) patients. The utilisation of activity trackers and the "Pace Me" app facilitated the quantification of energy levels, with participants instructed to maintain heart rate below 60% of maximum capacity for no more than 30 minutes per day. In cases of instances of permanent overruns, permanent symptoms or underruns, it was customary to make individual adjustments to the aforementioned limit. The objective of the intervention was to reduce the severity of post-exertional malaise (PEM). The study is a randomised controlled trial and includes 250 participants with LC. The results indicate that there is no difference in the severity of PEM between the intervention and standard care groups.

In general, the title is clear and communicates effectively that this is a randomised controlled trial. The paper is well structured and clearly formulated.

The summary offers a comprehensive overview of the study, including the rationale and specific aims. The introduction is clearly structured, addressing the scientific background and leading to the rationale for the study, with the specific aims clearly stated. The randomisation process and subsequent statistical analysis are both clear and satisfactory.

The study would have been enhanced by the inclusion of a parameter for daily mental workload, in addition to heart rate, to determine energy management limits. For instance, simple questions regarding mental exertion could have been evaluated using a Likert scale during the first week. This could be particularly beneficial given that PEM is characterised as physical and mental stress intolerance.

However, there are some issues, that should be addressed before publication of the manuscript.

Specific comments

Summary:

Lines 39-42:

Please note that the number of participants analysed for the intervention group does not match. Firstly, the six participants who withdrew from the intervention were subsequently omitted from the listing. Secondly, the number of analysed intervention participants should have been n=87. A comparison of the data with Figure 1 and the results is required, with any necessary corrections to the listing.

Intervention: 125 participants - 8 (lost to follow-up) - 24 participants (missing data) – 6 participants (discontinued intervention) = 87 participants

Lines 47-48:

The sentence "Digitally supported energy management in people with LC had no effect on PEM compared to standard care." is overly generalised and could be interpreted as applying to all LC patients and all digital methods of supporting energy management. Please clarify whether this result applies to your cohort, or rather, your digital support for energy management, the Pace Me app.

Introduction:

The section is well structured and presented.

Methods:

Randomisation and masking:

Line 154 Figure 1:

1. As outlined in the summary above, Figure 1 shows that of the 125 participants assigned to the intervention, eight were lost to follow-up and six discontinued the intervention, with the remaining 84 participants being evaluated. However, the correct number of intervention participants should be $n=87$.
2. It would be beneficial to ascertain the current status of the three participants not included in the listing. Could you kindly explain why these were not included in the statistical analysis?

Procedures:

Lines 164-167:

As you previously stated, the baseline questionnaires were completed during the online meeting, while the remaining questionnaires were completed via the app. Could you please confirm whether this is correct? Please can you confirm whether the questionnaires administered during the online meeting received active or passive support? It is important to note that active support during completion could potentially compromise the quality of the data, as the follow-up questionnaires had to be completed in a different format via the app. Could you please explain why the questionnaires were not also collected during a final online meeting? This would result in a standardised questionnaire survey. If this is not the case, please provide clearer wording in the text to make it more understandable.

Intervention Design:

Lines 187-188:

1. Please add the number of participants with unstable heart rate.
2. It would be important to know whether the participants with an unstable heart rate exceeded the limits of energy management more or less frequently than the other participants. The same applies to days with PEM or increased symptoms. Treacy et al. (2017) observed that wrist-worn activity trackers lose accuracy at speeds below 4.2 km/h. It has been demonstrated that the significance of the actual steps is underestimated as the pace slows down (<https://doi.org/10.1093/ptj/pzx010>). It is reasonable to assume that LC patients move around more slowly than healthy people in everyday life due to the restrictions caused by the various symptoms they experience, such as fatigue and pain. This suggests that these participants may have taken more steps than the activity tracker indicated. In this instance, it would be necessary to examine whether the transgression of the limits was perhaps not well recognised.
3. It would be interesting to know how you go about dealing with activities, such as cycling, that are not step countable.
4. In the event that this assumption were to be valid, it would be necessary to remove these participants from the analysis, as energy management in this form would be unfeasible for them.

Lines 193-196:

1. It is imperative to ascertain whether the participants who received the notification of 50% of the activity allowance during the afternoon or early evening were subsequently issued with further notifications, possibly even the 100% of the activity allowance. It is important to note that an inactive day could also have the opposite effect by specifying the percentages (50, 75, 100). For instance, the participants were encouraged to engage in slightly more physical activity.
1. Could you please clarify whether the participants were given any guidelines regarding energy management? For instance, were they encouraged to reach 75% as often as possible? If so, please add this to the text.

Results:

In general, the paper would benefit from improved characterisation of the participants. These could be included in the baseline characteristics of the supplementary information. This would include, for example, information on self-reported pre-existing medical conditions and the percentage of participants who were working, on sick leave, or unemployed during the study period. LC patients are a very heterogeneous group. During medical consultations with LC patients, it has been noted that symptoms may have been present prior to the onset of the condition, or may be explained by factors such as sleep apnoea or other pre-existing conditions. For those affected who are still in work, energy management and maintaining limits are significantly more difficult than for those on sick leave or unemployed, as work may involve fixed hours and non-self-determined workloads and activities. A more thorough characterisation could reveal points for discussion that could explain the lack of effect of the intervention.

Information regarding the percentage of participants who reached their individual limit per day (50%, 75%, 100%) would be advantageous.

Lines 246-252:

1. Please note that the number of participants analysed for the intervention group does not match. Firstly, the six participants who withdrew from the intervention were subsequently omitted from the listing. Secondly, the number of analysed intervention participants should have been $n=87$. A comparison of the data with Figure 1 and the summary is required, with any necessary corrections to the listing. However, the correct number of intervention participants should be $n=87$.
2. It would be beneficial to ascertain the current status of the three participants not included in the listing. Could you kindly explain why these were not included in the statistical analysis?

Line 271 table 2:

1. Please add an "I" to the "SF12 - Physical Health" field. Please ensure that you write consistently in rows 1 and 2 of the table for SF12.
2. Please check the Physical and Mental Health baseline and post-scores of the SF-12 for accuracy. It appears to be an anomalous finding that the mean values and the CI values for baseline and post-scores are identical for SF12 - Physical Health, with all participants having a score of 54 (CI 54-54). The same applies to SF12 - Mental Health. In this instance, the baseline and post-scores are also identical. However, a significant difference over time was reported, which appears

inconsistent with the values presented.

Discussion:

In general, the discussion was well-written and structured. However, in some places, it did become somewhat protracted.

Lines 276-278:

Please clarify whether this result applies to your cohort, or rather, your digital support for energy management.

The second paragraph:

1. This paragraph provides an overview of the recovery process for individuals affected by LC. This is also mentioned in the summary in lines 48-49. Could you please clarify whether there are any participants in your group who no longer have symptoms, or whose PEM screening is no longer positive?

2. The third section of the paper addresses the limitations of the study. It should also be noted that the participants' mental exertion was not taken into account in the energy management strategy. Managing mental exertion is a very central aspect of energy management. The definition of PEM (line 63) states that PEM refers to intolerance for physical and mental exertion. Assessing mental function using a monthly questionnaire via the app is not sufficient in this case. In future research, consideration should be given to the inclusion of mental exertion in energy management assessments.

Please could you also add this (2.) to the limitations in paragraph three.

Reviewer #3

(Remarks to the Author)

I have reviewed the manuscript entitled "A Digital Platform with Activity Tracking 1 for Energy Management Support in Long COVID: A Randomised Controlled Trial" with interest and in accordance with the journal's guidelines. This trial provides relevant insights into energy management for people with long COVID. However, I have identified several concerns with the methodology and analyses, which I believe merit further examination.

Main Issues

There are some inconsistencies in the flowchart figures that should be addressed to ensure transparency and accuracy in the reporting of patient flow. For instance, while 55 patients are reported as excluded, only 48 have a corresponding exclusion reason documented. Similarly, within the intervention group, there appears to be a discrepancy in the number of analyzed patients. Specifically, out of 125 patients who received the allocated intervention, 8 were lost to follow-up, 6 discontinued the intervention, and 24 were excluded from analysis due to incomplete post-intervention data. This should result in 87 patients remaining, yet the flowchart indicates a total of only 84.

The main analysis was conducted on participants who completed follow-up testing. However, no formal definition of the analysis population was provided (e.g. safety, intention-to-treat [ITT] or per-protocol [PP]). The authors should justify the population used for the primary analysis. Furthermore, repeating the analyses in an ITT population (i.e. all randomised participants), as is standard practice in clinical trials, would strengthen the validity of the findings.

A two-way mixed-model ANOVA was used in the item-level analysis (Table 1). However, this method may not be the most appropriate for item-level analysis involving Likert-type items scored from 0 to 4. This is mainly because ANOVA treats item scores as continuous with equal spacing between values, and the assumptions of normality and sphericity are difficult to justify with 5-point scales. Ordinal logistic regression with repeated measures or Generalised Estimating Equations for ordinal data would be a more appropriate method.

In conclusion, the authors claim that there were no adverse incidents and that the intervention was safe. However, they should explain in the methodology section how adverse and unintended events were defined (e.g., severity, timing) and assessed (e.g., spontaneous reporting, checklists, clinician assessment).

A baseline table summarising the demographic and clinical profile of the subjects included in the analysis, organised by study group, must be included as Table 1. This is standard practice when reporting on clinical trials and would help readers to understand the type of subjects included in the trial. The supplemental table showing the age and sex distribution is insufficient.

The authors used the CONSORT flowchart to describe subject inclusion and exclusion process. However, to ensure adherence to recognised standards for the transparent and complete clinical trial reporting, the authors should also include the 2025 CONSORT checklist as supplementary material.

Minor Issues

The summary should include the main study objective, as articulated at the end of the introduction, to provide readers with immediate context.

Post-hoc power calculations are generally discouraged, as they do not provide additional insight once results are known.

The authors should cite the exact R package (e.g. WebPower) used to estimate sample size, and also reference the R version and Jamovi platform used. The data sharing statement does not include any information on how to get in touch with the authors for a reasonable request.

The manuscript would benefit from the inclusion of a clear longitudinal plot showing each group's response to the DSQ-PEM questionnaire at both assessment points. Graphing the data will allow readers to visualize how activity tracking plus just-in-time messaging (intervention group) affected each outcome compared to standard care (control group). For instance, the numerical outcomes can be presented in the form of a graph that illustrates the evolution of each group's mean score at baseline and post. In the context of categorical outcomes, the utilization of a Sankey plot is a recommended approach for the graphical representation of alterations occurring at two timepoints.

The data sharing statement should include clear instructions on how to contact the corresponding author for reasonable data access requests

Reviewer #4

(Remarks to the Author)

Version 1:

Reviewer comments:

Reviewer #2

(Remarks to the Author)

I would like to express my sincere gratitude for your comprehensive response to the extensive comments.

In point 26 of the "Response to Referees Letter", a detailed explanation was provided, describing that the primary activity threshold was determined on the basis of heart rate.

Please accept my apologies for any inconvenience caused; it may be the case that I did not articulate my comment with sufficient precision.

The question regarding cycling should pertain to patients with unstable heart rates who were included in the analysis. The heart rate value could not be used as a benchmark in this case.

It is my understanding that such activity cannot be recorded and taken into account due to the unstable heart rate.

Consequently, the exclusion of these 23 patients from the analysis may be warranted, as their inclusion could potentially influence the results, for instance, if their cycling habits significantly impacted the outcomes.

Reviewer #3

(Remarks to the Author)

We would like to thank the authors for their detailed responses to the points we flagged in our review. Most of the answers are satisfactory.

The intention-to-treat analysis was reported in the Results section, under the subtitle: 'Sensitivity Analysis'. In randomized trials, the purpose of an ITT analysis is to estimate the effect of the intervention while maintaining the benefits of randomization, regardless of adherence to the protocol or any deviations from it. Sensitivity analyses, on the other hand, are additional analyses carried out to check the robustness of the main results to different assumptions. Therefore, the subtitle 'Sensitivity Analysis' must be removed or substituted with 'Intention-to-Treat Analysis'.

We appreciate the extended and detailed justification of the DSQ-PEM analysis, and we agree with the authors on most points. However, when reviewing the cited papers as examples of studies that have treated the DSQ-PEM as continuous data and analysed it using parametric tests such as ANOVA, we did not find the same approach if we understood correctly. Jason et al. (<https://www.mdpi.com/2035-8377/15/1/1>) and Nepotchatky et al. (<https://www.nature.com/articles/s41598-023-28955-9>) appear to report DSQ results as a 100-point composite score. In contrast, Twomey et al. (<https://pmc.ncbi.nlm.nih.gov/articles/PMC9383197/>) reported DSQ items using the step scoring method, where a frequency and severity score of 2,2 on any items 1–5 is considered indicative of PEM.

Reviewer #4

(Remarks to the Author)

REVIEWER COMMENTS

Reviewer #1 (Remarks to the Author):

1. This is a thoughtful well-written effort to test a mobile device for energy self-management in a pragmatic trial in people with Long COVID. Intervention guidance is based on self-report PEM in relation to objective heart rate and step counts. This guidance yielded Just-in-time messages to increase subject awareness of opportunities to adjust activity levels to reduce and better manage self-report PEM.

We thank R1 for their positive comments upon our manuscript.

2. The authors did not seek to determine at post-intervention if messages were actually perceived by subjects as “just in time,” an important element of usability. Given no significant improvement in the active intervention group despite fairly granular management feedback to subjects, post-intervention feedback from participants might have helped to identify potentially correctable issues with the protocol.

Thank you for your comment. We did seek post-intervention feedback from participants and these findings are reported elsewhere (<https://pubmed.ncbi.nlm.nih.gov/39529893/>). However, this feedback was centred around patient reported outcomes and how the intervention helped participants manage symptoms of LC. Although we did not focus on the timing of the messages per se, feedback from participants suggested that the PaceMe app was beneficial for (1) PEM management, (2) Support, (3) Validation, and (4) Control and Agency. This suggests that participants valued the intervention, even if it was not efficacious for reducing DSQ-PEM over six months and highlights the value of a digital health intervention as a vital component of rehabilitation.

3. Also, data about psychological and affective states (motivation, mood, stress levels) which can influence physical activity, sedentary behavior and PEM were not collected. For instance, patients with ME/CFS and Long COVID may disregard symptom-based signals (e.g., increased headache) of upcoming crashes, a manifestation of severe PEM. Ignoring such signals may be driven by self-defeating motivation to stay as active as possible in the moment and wishful thinking about avoiding a subsequent crash. Identifying these affective states during an intervention may increase patient awareness of how they can and do undermine energy self-management.

Thank you for raising this, and we fully agree with your point. Unfortunately, understanding the impact of affective states was not a focus of this study and these data were not collected. At the time of developing and rolling out this pragmatic trial, the overarching goal was to provide people with LC with a potential solution that would help them manage their energy levels and prevent PEM. It required a careful balance of gathering enough data to answer essential research questions but not too much that would impose undue burden on participants, especially considering the nature of the topic. We involved PPI groups throughout the planning and execution of the trial to ensure the needs of the target group were met. But we acknowledge that not all potential mediators or moderators were evaluated. We have added a statement about this to our limitations.

“Thirdly, we did not assess participants’ psychological or affective states (e.g., motivation, mood, stress), which may influence physical activity behaviour, pacing decisions, and susceptibility to post-exertional malaise.”

4. No adherence data were presented. Although “guidance” rather than direct instructions to modify activity levels was provided, it is unclear how often the guidance offered was actually followed. For instance, was a guidance message that activity level was too high

followed by a reduction in steps? If such guidance was ignored, this also suggests the importance of collaborative sessions with study subjects regarding study design.

Thank you for raising this, and we fully agree with your point. Unfortunately, we feel this is outside the scope of the present manuscript. Our primary aim was to measure effectiveness of the intervention, regardless of engagement/adherence/compliance etc. Adherence/compliance data would be indicative of measuring behavioural change, which is of course interesting, but was not our primary aim. We may explore this suggestion in a future manuscript but consider it exploratory analysis rather than answering our primary research question. The complication is also that individuals could receive four messages per day (25%, 50%, 75%, 100% energy allowance used), and we would need to set a proximity to the message being sent to the window of behaviour change (i.e. HR reduced for 1, 2, 3, 4, 5, $n??$ hours). There are many decisions which we would need to rationalise before we even consider writing the script to pull down this data.

However, we have included the number of 100% received on average in the results section, which may be indicative of energy allowance exceeded, which may indicate adherence.

“The mean and standard deviation (\pm) number of 100% messages received per participant in month 1, 2, 3, 4, 5, and 6 was 13 ± 10 , 10 ± 10 , 8 ± 9 , 7 ± 9 , 6 ± 10 , and 6 ± 10 , respectively. The mean number of days those participants exceeded their 100% energy allowance over the full six months was 51 ± 50 days.”

5. Although Energy Management is the central construct and intervention goal, it is not clear how energy was measured. No definition was given. The authors noted that Energy Management requires participants to recall and compare two poorly described constructs; 'energy availability' and 'energy use', especially problematic when dealing with impaired patients. It seems that energy was used as more of an appealing intervention label rather than an operationalized construct. This issue should be addressed.

We thank R1 for this observation and apologise for the ambiguity in prose. Energy management is indeed a difficult construct to define. We imagine this is the reason some have resorted to metaphors such as the spoon theory (<https://www.rcot.co.uk/latest-news/spoon-theory>). We have provided the definition in the introduction: “Energy management uses techniques such as planning, feedback, and activity diaries to help individuals manage their ‘energy envelope’ to limit the debilitating consequences of their condition by reducing or avoiding periods of symptom exacerbation.”

Likewise, ‘energy availability’ and ‘energy used’ are vague constructs, which we argue are immeasurable in the current context. Of course, the SI unit of energy is the joule (J), sometimes expressed in kcal for the consumer. It would be possible for us to calculate energy expenditure from heart rate (<https://journals.sagepub.com/doi/abs/10.1260/174795407782233146>), but this would be moving one degree away from the unit we measured.

Our initial limit was set based on guidance from Physios for ME (one of the authors on this manuscript). The details of the algorithm are outlined in the methods of the manuscript: “The intervention used a personalised activity allowance, initially set to no more than 30 minutes of activity above 60% of their age-predicted heart rate maximum. This allowance was iteratively adjusted in response to participants’ activity and PEM reports. If participants reported PEM and had exceeded the allowance, the allowance remained unchanged (i.e. the PEM may be due to poor pacing). If participants reported PEM but had remained within their allowance for the preceding 3 days, the allowance was reduced in a stepwise fashion (i.e. activity allowance may be too large to prevent

PEM). Conversely, the allowance was increased if participants did not report PEM for three consecutive weeks.” This threshold is considered the starting point that Physios for ME use to aid people with ME/CFS integrate HR pacing into their lives. At the time of this project inception, there were no data on pacing in LC, so we used the limits commonly used in ME/CFS.

To operationalise energy management, we used HR thresholds as described above, in an attempt to reduce cognitive burden of manually tracking activities and PEM. We have included the following image in supplementary material to outline the data pipeline. We can move this into the manuscript if you feel this would explain the process better to the reader.

- There may be no published peer review data on the use of digital technology for energy management in CFS, but the commercial company Visible has collected apparently enormous amounts of mobile HR/HRV and activity monitor data in Long COVID and CFS patients in order to generate pacing suggestions via their wearable app. It might inform the authors' thinking about their mobile app trial to communicate with this company. A recently posted paper apparently based on Visible technology data on Research Square (Aitken et al; DOI: 10.21203/rs.3.rs-5423422/v1) provides predictive models of HR and HRV in relation to symptom exacerbations in ME/CFS and Long COVID using data from mobile device technology.

We thank R1 for their suggestion. We have added the following paragraph to the introduction. We hope this expands upon the current state of energy management knowledge in long COVID.

Despite its widespread use in ME/CFS, evidence to support the efficacy of energy management in preventing bouts of PEM is extremely limited^{11,15}. Encouragingly, Parker and colleagues¹⁶ provided preliminary evidence that a structured energy management protocol significantly reduced the number of PEM episodes in people with LC. However, upon closer inspection, this intervention seems analogous to graded exercise therapy (a gradual increase in physical activity over time), rather than energy management *per se*, which is highly contentious with the LC and ME/CFS support groups. To our knowledge, no contemporary efficacy trials have leveraged mobile and wearable technology to address the challenges of tracking activity to manage LC symptoms^{11,15}. A protocol for a feasibility study including heart rate variability biofeedback in

people with LC was published in 2022¹⁷, with the feasibility study published in 2024¹⁸. However, only 13 participants completed the study, limiting confidence in results. On the other hand, a commercially available platform (Visible), has already launched biofeedback-informed energy management support, backed by a reported \$1m in start-up funding and over 100,000 users. According to the company's recent preprint, within-person changes in HRV and heart rate emerged as predictors of symptom change, with higher heart rate and lower HRV conferring risk for 'crashes', 'fatigue', [OBJ/OBJ]. However, we argue there are several limitations with this platform that render it unsuitable for research purposes. Firstly, the concepts of 'crashes', 'fatigue', and 'brain fog' are quantified without the use of validated instruments, but a 1-3 rating system. This means that heart rate and HRV may predict 1, 2, or 3, but whether 1, 2, or 3 is a valid representation of 'crash' (i.e. PEM) is uncertain. Secondly, as this is a consumer-focussed application, it is behind a paywall which creates bias in 'recruitment' (they are arguably consumers rather than participants). Thirdly, the HRV and heart rate data were collapsed into large time epochs, which is necessary for data analysis due to the size of the dataset, but this does not permit in-the-moment warnings of upcoming PEM (or 'crash'). Finally, it suffers from ascertainment bias in that participants self-declare their impairment and as such, the dataset contains people with ME/CFS, Long COVID, and any other co-morbidities, anyone feels tired often, and anyone who downloads it for interest. Anyone can contribute to the Visible dataset. Currently, there is no platform that warns participants of potential PEM, informed by minute-by-minute biofeedback, using validated instruments, without a subscription fee.

7. Detail

Line 65; myalgic encephalitis—misspelled. Should be: Encephalomyelitis
Corrected as suggested.

8. Line 103. The trial ran between January 2022 and September 2023. But the paper later states: At the time of recruitment, the UK was engaged in COVID-19 related lockdowns. Please clarify. Lockdowns ended in 2020.

We thank Reviewer 1 for this observation. The statement has been removed, and the reference to recruitment during lockdown has been corrected. The revised text now reads: 'The trial ran between January 2022 and September 2023, with recruitment taking place from November 2021 to the end of February 2023.'

9. What are the noteworthy results?

This is an initial controlled trial of mobile digital technology consisting of age-predicted heart rate maximum, step counts and PEM reporting to help guide and manage activity levels that maximize physical function and moderate risk of triggering PEM.

We thank R1 for their positive comments upon our manuscript.

10. Will the work be of significance to the field and related fields? How does it compare to the established literature? If the work is not original, please provide relevant references. The trial does break new ground in the sense that no previously published trials have used this algorithmic approach in Long COVID or ME/CFS. The intervention combines objective measures and symptom self-report to improve illness management all housed in a wearable device. Although there is no published scientific literature on this, a popular commercial device, i.e., Visible, takes a very similar approach to Long COVID and ME/CFS self-management. A reference to this product would indicate a more thorough review of such devices.

We thank R1 for their positive comments upon our manuscript and have referred to the Visible app in the introduction as requested.

11. Does the work support the conclusions and claims, or is additional evidence needed?
The data does support its conclusions which are appropriately modest given null findings.

Thank you.

12. Are there any flaws in the data analysis, interpretation and conclusions? - Do these prohibit publication or require revision?
Revision is required to more thoroughly consider the methodological limitations of the study as indicated above.

We thank R1 for these comments and hope we have addressed your comments satisfactorily.

13. Is the methodology sound? Does the work meet the expected standards in your field?
The methodology is consistent with the standards of a pragmatic RCT.

Thank you.

14. Is there enough detail provided in the methods for the work to be reproduced?
Yes. The methodology contains sufficient detail to be reproduced. Given the technicals of a device-based intervention some consultation would probably be necessary.

Thank you. We have included large supplementary material to outline the technical aspects of the intervention, but including this within the manuscript would exceed the word limit of *Nature Comms*.

15. **Reviewer #2 (Remarks to the Author):**

General comments

As a reviewer, I had the privilege of reviewing the paper "A Digital Platform with Activity Tracking for Energy Management Support in Long COVID: A Randomized Controlled Trial".

We thank R2 for their positive comment about our manuscript.

16. The present study examines the effects of a 6-month just-in-time intervention to support energy management in Long COVID (LC) patients. The utilisation of activity trackers and the "Pace Me" app facilitated the quantification of energy levels, with participants instructed to maintain heart rate below 60% of maximum capacity for no more than 30 minutes per day. In cases of instances of permanent overruns, permanent symptoms or underruns, it was customary to make individual adjustments to the aforementioned limit. The objective of the intervention was to reduce the severity of post-exertional malaise (PEM). The study is a randomised controlled trial and includes 250 participants with LC. The results indicate that there is no difference in the severity of PEM between the intervention and standard care groups. In general, the title is clear and communicates effectively that this is a randomised controlled trial. The paper is well structured and clearly formulated.

We thank R2 for their positive comments about our manuscript, and it is reassuring that R2 has grasped the overall methodology of the project, as it is challenging to communicate.

17. The summary offers a comprehensive overview of the study, including the rationale and specific aims. The introduction is clearly structured, addressing the scientific background

and leading to the rationale for the study, with the specific aims clearly stated. The randomisation process and subsequent statistical analysis are both clear and satisfactory.

We thank R2 for their positive comments about our manuscript.

18. The study would have been enhanced by the inclusion of a parameter for daily mental workload, in addition to heart rate, to determine energy management limits. For instance, simple questions regarding mental exertion could have been evaluated using a Likert scale during the first week. This could be particularly beneficial given that PEM is characterised as physical and mental stress intolerance.

We thank R2 for their comment, but these data were not collected at the time. We agree it would be useful to determine cognitive load, as well as heart rate, but this was outside the scope of the project. Our aim was to assess HR supported activity management and we agree that adding mental workload would have been useful but there is no passive way of such data collection, thus requiring daily updates from users which in turn involves daily prompts, reminders, and notifications, all adding to participant burden. We have included some text in the limitations to address this:

“...participants' mental exertion was not considered in the energy management strategy. Managing mental exertion is of course a central aspect of energy management. In future research, consideration should be given to the inclusion of mental exertion in energy management assessments, although this could increase participant burden so a passive measure of mental exertion would be beneficial.”

19. However, there are some issues, that should be addressed before publication of the manuscript.

Specific comments

Summary:

Lines 39-42:

Please note that the number of participants analysed for the intervention group does not match. Firstly, the six participants who withdrew from the intervention were subsequently omitted from the listing. Secondly, the number of analysed intervention participants should have been $n=87$. A comparison of the data with Figure 1 and the results is required, with any necessary corrections to the listing.

Intervention: 125 participants - 8 (lost to follow-up) - 24 participants (missing data) – 6 participants (discontinued intervention) = 87 participants

We apologise for this oversight. We made an error in the text/figure. The number of participants analysed ($n=84$) was correct, and we have amended the text in the abstract, results, and figure 1 (CONSORT).

20. Lines 47-48:

The sentence "Digitally supported energy management in people with LC had no effect on PEM compared to standard care." is overly generalised and could be interpreted as applying to all LC patients and all digital methods of supporting energy management. Please clarify whether this result applies to your cohort, or rather, your digital support for energy management, the Pace Me app.

We have amended this to:

“In this study, energy management supported with the Pace Me app had no effect on PEM in people with LC compared to standard care.”

We hope this improves the clarity of the abstract.

21. Introduction:

The section is well structured and presented.

We thank R2 for their positive comments about our introduction.

22. Methods:

Randomisation and masking:

Line 154 Figure 1:

As outlined in the summary above, Figure 1 shows that of the 125 participants assigned to the intervention, eight were lost to follow-up and six discontinued the intervention, with the remaining 84 participants being evaluated. However, the correct number of intervention participants should be n=87.

We apologise for this oversight. We made an error in the text/figure. The number of participants analysed (n=84) was correct, and we have amended the text in the abstract, results, and figure 1 (CONSORT).

23. It would be beneficial to ascertain the current status of the three participants not included in the listing. Could you kindly explain why these were not included in the statistical analysis?

We apologise for this oversight. We made an error in the text/figure. The number of participants analysed (n=84) was correct, and we have amended the text in the abstract, results, and figure 1 (CONSORT).

24. Procedures:

Lines 164-167:

As you previously stated, the baseline questionnaires were completed during the online meeting, while the remaining questionnaires were completed via the app. Could you please confirm whether this is correct?

We have now corrected this, 'During the meeting, participants were encouraged to complete all baseline questionnaires afterward to avoid adding to their burden, as the meeting itself lasted an hour.'

25. Please can you confirm whether the questionnaires administered during the online meeting received active or passive support? It is important to note that active support during completion could potentially compromise the quality of the data, as the follow-up questionnaires had to be completed in a different format via the app. Could you please explain why the questionnaires were not also collected during a final online meeting? This would result in a standardised questionnaire survey. If this is not the case, please provide clearer wording in the text to make it more understandable.

We have now included a section that reads:

"At meeting 2, participants were encouraged to fill in the questionnaires as soon after the meeting as possible. We checked the database one week later and contacted them if they had not completed the questionnaires to ensure baseline data capture before starting the intervention or control part of the experiment."

"Participants were contacted at the end of the study to complete all questionnaires before returning their Fitbit or being provided with the energy management platform as a thank you for participation"

Intervention Design:

Lines 187-188:

1. Please add the number of participants with unstable heart rate.

We have now included a section that reads: " Throughout the study, thirty-three participants reported unstable heart rate data and were switched to step count monitoring. In the final per-protocol analysis, this applied to 23 participants. These limits were not intended as targets but as a guide for energy management."

26. It would be important to know whether the participants with an unstable heart rate exceeded the limits of energy management more or less frequently than the other participants. The same applies to days with PEM or increased symptoms. Treacy et al. (2017) observed that wrist-worn activity trackers lose accuracy at speeds below 4.2 km/h. It has been demonstrated that the significance of the actual steps is underestimated as the pace slows down (<https://doi.org/10.1093/ptj/pzx010>). It is reasonable to assume that LC patients move around more slowly than healthy people in everyday life due to the restrictions caused by the various symptoms they experience, such as fatigue and pain. This suggests that these participants may have taken more steps than the activity tracker indicated. In this instance, it would be necessary to examine whether the transgression of the limits was perhaps not well recognised. It would be interesting to know how you go about dealing with activities, such as cycling, that are not step countable.

In the event that this assumption were to be valid, it would be necessary to remove these participants from the analysis, as energy management in this form would be unfeasible for them.

Thank you for your thoughtful comments regarding the limitations of wrist-worn activity trackers in detecting slow and non-ambulatory movement, and the implications for energy management in individuals with Long COVID.

We would like to clarify that our intervention was primarily based on heart rate, not step count. Specifically, participants received messaging triggered by exceeding a personalised heart rate allowance, which was defined as no more than 30 minutes of activity above 60% of their age-predicted heart rate maximum. For participants with unstable heart rates, step counts were used to set activity allowances as an alternative. During the study N=23 (per-protocol) participants were moved to step-based thresholds on request. For those step thresholds, we acknowledge that step-based thresholds may underestimate activity in slow movers, as noted in Treacy et al. (2017).

By basing the primary activity threshold on heart rate rather than steps, we aimed to ensure that slow walking and non-ambulatory activities (e.g., cycling) were still captured, as these activities typically result in an elevated heart rate even when step counts are low or absent. For cycling and other non-step activities, the Fitbit Charge 5 records heart rate continuously, allowing us to monitor exertion regardless of step count. Previous studies using the Fitbit Charge HR (an earlier model with similar sensor technology to the Charge 5) generally demonstrated strong to very strong correlations with the reference standard (polar chest strap) in free living conditions, with mean absolute percentage errors typically within $\pm 10\%$ which is considered acceptable for consumer-grade devices (Fuller D, Colwell E, Low J, Orychock K, Tobin MA, Simango B, Buote R, Van Heerden D, Luan H, Cullen K, Slade L, Taylor NGA. Reliability and Validity of Commercially Available Wearable Devices for Measuring Steps, Energy Expenditure, and Heart Rate: Systematic Review. JMIR Mhealth Uhealth. 2020 Sep 8;8(9):e18694. doi: 10.2196/18694). However, the device consistently underestimated HR, although the degree of underestimation increases at higher activity intensities. For example, in adults, the underestimation was greater during moderate-to-vigorous activity, but group-level errors still fell within 10%. We have added the following to the limitations section:

“While Fitbit devices have demonstrated reasonable validity for step counting and heart rate tracking in healthy adults, their accuracy diminishes at slower walking speeds Treacy et al. (2017) and higher intensity activity (Fuller et al 2020), respectively. Therefore, some episodes of overexertion may not have been adequately detected.”

27. Lines 193-196:

It is imperative to ascertain whether the participants who received the notification of 50% of the activity allowance during the afternoon or early evening were subsequently issued with further notifications, possibly even the 100% of the activity allowance. It is important to note that an inactive day could also have the opposite effect by specifying the percentages (50, 75, 100). For instance, the participants were encouraged to engage in slightly more physical activity.

We thank the reviewer for this suggestion. We can confirm that participants who received the 50% notification may be subsequently issued with further notifications, possibly even the 100% notification if they exceeded their allowance.

We can clarify that participants were never encouraged to engage in activity that would raise heart rate (see the next response to reviewer also). They were only ever alerted when they were at risk of doing too much activity. We have included this section: "During the demonstration of the app and pacing feature, participants were informed that if they did not receive a notification indicating they had reached 50% of their daily activity allowance, they were on track. Pacing notifications were only sent to prompt users to slow down. This understanding was evident in the emails we received, in which participants apologised for overexerting themselves when they received a 100% notification." Moreover, we have included the number of 100% messages received by the intervention group in the methods section:

“The mean and standard deviation (\pm) number of 100% messages received per participant in month 1, 2, 3, 4, 5, and 6 was 13 ± 10 , 10 ± 10 , 8 ± 9 , 7 ± 9 , 6 ± 10 , and 6 ± 10 , respectively. The mean number of days those participants exceeded their 100% energy allowance over the full six months was 51 ± 50 days.”

28. Could you please clarify whether the participants were given any guidelines regarding energy management? For instance, were they encouraged to reach 75% as often as possible? If so, please add this to the text.

We thank the reviewer for this suggestion. We have included this section: "During the demonstration of the app and pacing feature, participants were informed that if they did not receive a notification indicating they had reached 50% of their daily activity allowance, they were on track. Pacing notifications were only sent to prompt users to slow down. This understanding was evident in the emails we received, in which participants apologised for overexerting themselves when they received a 100% notification."

29. Results:

In general, the paper would benefit from improved characterisation of the participants. These could be included in the baseline characteristics of the supplementary information. This would include, for example, information on self-reported pre-existing medical conditions and the percentage of participants who were working, on sick leave, or unemployed during the study period. LC patients are a very heterogeneous group. During medical consultations with LC patients, it has been noted that symptoms may have been present prior to the onset of the condition, or may be explained by factors such as sleep apnoea or other pre-existing conditions. For those affected who are still in

work, energy management and maintaining limits are significantly more difficult than for those on sick leave or unemployed, as work may involve fixed hours and non-self-determined workloads and activities. A more thorough characterisation could reveal points for discussion that could explain the lack of effect of the intervention.

We thank the reviewer for this feedback. While we did not collect any pre-existing medical data, but participants with other pre-existing fatigue-related conditions were excluded from the study. We have updated the inclusion and exclusion criteria section of the manuscript to make this clearer. We have included a supplementary table to show the gender and age distribution between groups, and have included a supplementary figure to visualise the change to employment caused by LC at baseline.

30. Information regarding the percentage of participants who reached their individual limit per day (50%, 75%, 100%) would be advantageous.

We have included the number of 100% messages received by the intervention group by month in the methods section:

“The mean and standard deviation (\pm) number of 100% messages received per participant in month 1, 2, 3, 4, 5, and 6 was 13 ± 10 , 10 ± 10 , 8 ± 9 , 7 ± 9 , 6 ± 10 , and 6 ± 10 , respectively. The mean number of days those participants exceeded their 100% energy allowance over the full six months was 51 ± 50 days.”

31. Lines 246-252:

Please note that the number of participants analysed for the intervention group does not match. Firstly, the six participants who withdrew from the intervention were subsequently omitted from the listing. Secondly, the number of analysed intervention participants should have been $n=87$. A comparison of the data with Figure 1 and the summary is required, with any necessary corrections to the listing. However, the correct number of intervention participants should be $n=87$.

We apologise for this oversight. We made an error in the missing data text/figure. The number of participants analysed ($n=84$) was correct, and we have amended the text in the abstract, results, and figure 1 (CONSORT).

32. It would be beneficial to ascertain the current status of the three participants not included in the listing. Could you kindly explain why these were not included in the statistical analysis?

We apologise for this oversight. We made an error in the missing data text/figure. The number of participants analysed ($n=84$) was correct, and we have amended the text in the abstract, results, and figure 1 (CONSORT).

33. Line 271 table 2:

1. Please add an "l" to the "SF12 - Physical Health" field. Please ensure that you write consistently in rows 1 and 2 of the table for SF12.

Corrected as suggested. We have added the 'l' to health and capitalised the H for consistency.

34. Please check the Physical and Mental Health baseline and post-scores of the SF-12 for accuracy. It appears to be an anomalous finding that the mean values and the CI values for baseline and post-scores are identical for SF12 - Physical Health, with all participants having a score of 54 (CI 54-54). The same applies to SF12 – Mental Health. In this

instance, the baseline and post-scores are also identical. However, a significant difference over time was reported, which appears inconsistent with the values presented. Thank you for this observation. We have previously checked and rechecked these statistics because we thought it implausible. However, it is worth noting that the SF-12 conversion (for comparison with the SF-36) has some log functions which reduces the variability.

Below are the results for the SF-12 physical (pulled straight from Jamovi).

Within Subjects Effects

	Sum of Squares	df	Mean Square	F	p
time	0.0159	1	0.0159	0.0986	0.754
time * group	0.1424	1	0.1424	0.8843	0.349

Below are the results for the SF-12 mental health (pulled straight from Jamovi).

Within Subjects Effects

	Sum of Squares	df	Mean Square	F	p
time	2.950	1	2.950	9.552	0.003
time * group (2)	0.122	1	0.122	0.396	0.531

We made a minor error in rounding in table 2, which has now been corrected. Control pre: 59 (59-60), control post: 59 (59-60). Intervention pre: 59 (59-59), intervention post: 59 (59-59).

TO

Control pre : 59 (59-59), control post: 59 (59-60). Intervention pre: 59 (59-59), intervention post: 59 (59-60).

Still, we are surprised with the p value.

38. Discussion:

In general, the discussion was well-written and structured. However, in some places, it did become somewhat protracted.

Lines 276-278:

Please clarify whether this result applies to your cohort, or rather, your digital support for energy management.

Apologies, we have amended this to state 'the present study was...'

39. The second paragraph:

This paragraph provides an overview of the recovery process for individuals affected by LC. This is also mentioned in the summary in lines 48-49. Could you please clarify whether there are any participants in your group who no longer have symptoms, or whose PEM screening is no longer positive?

We thank the reviewer for this comment and have assumed they mean the DSQ-PEM screening criteria of ≥ 2 for both frequency and severity on a specific item, which suggests PEM is both present and clinically relevant. We have included the following analyses in the results section:

In terms of screening for PEM, a score of ≥ 2 for both frequency and severity on a specific item (Q1-5) suggests PEM is both present and clinically relevant. Therefore, participants could be either positive or negative for PEM. When groups were combined, McNemar's Test revealed the number of individuals classified as PEM positive decreased from baseline to month 6 ($p=0.004$). 111 participants were considered PEM positive at baseline and month 6, 31 participants were PEM positive at baseline but not at follow-up (improvement), 12 participants were PEM negative before but positive after (worsening), and 7 participants were PEM negative at both time points. In the intervention group, McNemar's Test revealed no change in PEM status from baseline to month 6 ($p=0.394$). 59 participants were considered PEM positive at baseline and month 6, 13 participants were PEM positive at baseline but not at follow-up (improvement), 9 participants were PEM negative before but positive after (worsening), and 3 participants were PEM negative at both time points. In the control group, the number of individuals classified as PEM positive decreased from baseline to month 6 ($p=0.001$). 52 participants were considered PEM positive at baseline and month 6, 18 participants were PEM positive at baseline but not at follow-up (improvement), 3 participants were PEM negative before but positive after (worsening), and 4 participants were PEM negative at both time points. Indeed, 19 improved and five worsened. We subsequently used Chi squared test for between group effects which exhibited no differences at baseline ($p=0.411$) or month 6 ($p=0.155$).

40. The third section of the paper addresses the limitations of the study. It should also be noted that the participants' mental exertion was not taken into account in the energy management strategy. Managing mental exertion is a very central aspect of energy management. The definition of PEM (line 63) states that PEM refers to intolerance for physical and mental exertion. Assessing mental function using a monthly questionnaire via the app is not sufficient in this case. In future research, consideration should be given to the inclusion of mental exertion in energy management assessments. Please could you also add this (2.) to the limitations in paragraph three.

We thank R2 for this suggestion and have added the following to the discussion:

“Secondly, participants' mental exertion was not considered in the energy management strategy. Managing mental exertion is of course a central aspect of energy management. In future research, consideration should be given to the inclusion of mental exertion in energy management assessments, although this could increase participant burden so a passive measure of mental exertion would be very beneficial.”

41. **Reviewer #3 (Remarks to the Author):**

I have reviewed the manuscript entitled "A Digital Platform with Activity Tracking 1 for Energy Management Support in Long COVID: A Randomised Controlled Trial" with interest and in accordance with the journal's guidelines. This trials provides relevant insights into energy management for people with long COVID. However, I have identified several concerns with the methodology and analyses, which I believe merit further examination.

We thank R3 for their positive comments about our manuscript.

42. Main Issues

There are some inconsistencies in the flowchart figures that should be addressed to ensure transparency and accuracy in the reporting of patient flow. For instance, while 55 patients are reported as excluded, only 48 have a corresponding exclusion reason documented. Similarly, within the intervention group, there appears to be a discrepancy in the number of analyzed patients. Specifically, out of 125 patients who received the allocated intervention, 8 were lost to follow-up, 6 discontinued the intervention, and 24 were excluded from analysis due to incomplete post-intervention data. This should result in 87 patients remaining, yet the flowchart indicates a total of only 84.

We apologise for this oversight. We made an error in the missing data text/figure. The number of participants analysed (n=84) was correct, and we have amended the text in the abstract, results, and figure 1 (CONSORT).

43. The main analysis was conducted on participants who completed follow-up testing. However, no formal definition of the analysis population was provided (e.g. safety, intention-to-treat [ITT] or per-protocol [PP]). The authors should justify the population used for the primary analysis. Furthermore, repeating the analyses in an ITT population (i.e. all randomised participants), as is standard practice in clinical trials, would strengthen the validity of the findings.

We thank the reviewer for this suggestion. In the initial submission, we did convey that a per protocol analysis would be conducted. However, we agree with the reviewer that repeating the analyses in an ITT population (i.e. all randomised participants), has strengthened the validity of the findings.

Along with published guidance (<https://www.bmj.com/content/342/bmj.d40>), we have added ITT analysis to confirm our findings were robust. The statistical analysis section and results section now reflects this. Broadly speaking, the results from the ITT support the PP analysis, with two p values moving from just over/under 0.05 to just under/over 0.05. However, no wholesale differences were observed, so we are confident in our conclusions. Thank you again for this suggestion. We feel the change had added robustness to the manuscript.

44. A two-way mixed-model ANOVA was used in the item-level analysis (Table 1). However, this method may not be the most appropriate for item-level analysis involving Likert-type items scored from 0 to 4. This is mainly because ANOVA treats item scores as continuous with equal spacing between values, and the assumptions of normality and sphericity are difficult to justify with 5-point scales. Ordinal logistic regression with repeated measures or Generalised Estimating Equations for ordinal data would be a more appropriate method.

We appreciate R3's concern regarding the treatment of the 5-point Likert scale as continuous data. While Likert-type items are sometimes considered ordinal in nature, there is substantial scholarly support for treating aggregated or multi-point Likert scales as continuous variables. Our methodological decision aligns with established psychometric practices and empirical evidence. Indeed, the initial development and testing of the DSQ-PEM treated questions 1-5 as continuous data (<https://pmc.ncbi.nlm.nih.gov/articles/PMC4788471/>), and thus reported means and SDs. The typical criticism of this approach is that difference between responses as not considered equal. However, for the DSQ-PEM frequency responses, 0 is none of the

time, 4 is all of the time, and 2 is 'about half of the time', indicating that 4 is indeed double 2, and 0 is a true zero.

Indeed, several other authors have treated the DSQ-PEM as continuous data and analysed these data using parametric tests such as ANOVA

(<https://www.mdpi.com/2035-8377/15/1/1>; <https://www.nature.com/articles/s41598-023-28955-9>; <https://pmc.ncbi.nlm.nih.gov/articles/PMC9383197/>).

Additionally, when multiple items are summed or averaged to form a scale—particularly with five or more categories—researchers often treat the resulting variable as approximately continuous. This is predicated on the assumption that measurement error and individual item nonlinearity are mitigated through aggregation

(<https://link.springer.com/article/10.1007/s10459-010-9222-y>). Empirical and simulation-based studies have demonstrated that parametric statistical methods, including linear regression and ANOVA, yield robust results when applied to Likert-type data with five or more response categories. For instance, Norman

(<https://link.springer.com/article/10.1007/s10459-010-9222-y>) argues that the central limit theorem supports the use of parametric techniques, particularly as the number of items and respondents increases. Similarly, Carifio and Perla

(<https://pubmed.ncbi.nlm.nih.gov/19120943/>) contend that treating Likert scales as interval-level data is both methodologically and practically justifiable, especially when the scales are symmetrical and balanced (as per the DSQ-PEM).

Moreover, the psychometric literature supports the notion that the precision and informational value of Likert scales improve with more response options. Harpe

(<https://www.sciencedirect.com/science/article/abs/pii/S1877129715200196>)

emphasises that Likert scales with five or more points (as per the DSQ-PEM) approach interval-level measurement properties, thus allowing for their treatment as continuous variables in many applied settings. In practice, many prominent psychological and social science instruments (e.g., Beck Depression Inventory, Rosenberg Self-Esteem Scale) employ Likert-style formats and are analysed using parametric methods, further supporting the practical viability of this approach.

In the present study, the 0–4 scale was used consistently across multiple items, with responses averaged to generate the composite score as our primary outcome. The DSQ-PEM exhibited high consistency (Test-retest: Pearson's correlation coefficients = 0.85 or higher. Internal consistency: Cronbach's alpha = 0.88

[<https://pmc.ncbi.nlm.nih.gov/articles/PMC4871625/>] and 0.95

[<https://pmc.ncbi.nlm.nih.gov/articles/PMC4830389/>]), suggesting the items formed a coherent and reliable construct. Treating the resulting scale as continuous allowed us to leverage the benefits of parametric modelling, including increased statistical power and interpretability.

While we acknowledge the theoretical distinction between ordinal and interval measurement levels, we assert that our analytic approach is consistent with best practices in applied quantitative research, and other researchers using the DSQ-PEM.

45. In conclusion, the authors claim that there were no adverse incidents and that the intervention was safe. However, they should explain in the methodology section how adverse and unintended events were defined (e.g., severity, timing) and assessed (e.g., spontaneous reporting, checklists, clinician assessment).

We thank the reviewer for this comment and have added the following to the manuscript:

Within the app, there was a 'request a call back' function, and it was explained this was how participants could report adverse effects.

46. A baseline table summarising the demographic and clinical profile of the subjects included in the analysis, organised by study group, must be included as Table 1. This is standard practice when reporting on clinical trials and would help readers to understand the type of subjects included in the trial. The supplemental table showing the age and sex distribution is insufficient.

We thank the reviewer for this feedback but argue this is not a typical clinical trial. We did not collect any pre-existing medical data, participants with other pre-existing fatigue-related conditions were excluded from the study. We have ensured the inclusion and exclusion criteria section of the manuscript is clearer. Additionally, we have also added a section on self-reported occupation status at baseline to supplementary material.

47. The authors used the CONSORT flowchart to describe subject inclusion and exclusion process. However, to ensure adherence to recognised standards for the transparent and complete clinical trial reporting, the authors should also include the 2025 CONSORT checklist as supplementary material.

Added as requested.

48. Minor Issues

The summary should include the main study objective, as articulated at the end of the introduction, to provide readers with immediate context.

We thank the reviewer for this suggestion and have added the following to the start of the discussion: “The aim of this trial was to determine if activity tracking combined with just-in-time messages helped people with LC implement energy management and reduce incidents of symptom exacerbation. We hypothesised a priori that a just-in-time intervention to assist energy management (intervention group) would reduce frequency or severity of PEM compared to usual care (control). The main finding of the present study was that ...”

49. Post-hoc power calculations are generally discouraged, as they do not provide additional insight once results are known.

We have now removed this section.

50. The authors should cite the exact R package (e.g. WebPower) used to estimate sample size, and also reference the R version and Jamovi platform used. The data sharing statement does not include any information on how to get in touch with the authors for a reasonable request.

We thank R3 for this suggestion and have now added the following:

“Using the WebPower package in R Studio (version 2024.04.2+764), and the `wp.rmanova` function, with two groups, two time points, $f=0.25$, assuming sphericity, $\alpha=0.05$, $1-\beta=0.9$, testing for an interaction effect, the total n was 170 (85 per group). Consequently, we aimed to recruit 125 participants per group to allow for 30% drop-out. All analyses were conducted using Jamovi version 2.3.21.”

Additionally, the data sharing agreement has been updated:

“Data collected for this study, including individual anonymised participant data and a data dictionary defining each field in the set will be made available to others upon reasonable request to the corresponding author (l.hayes4@lancaster.ac.uk).”

51. The manuscript would benefit from the inclusion of a clear longitudinal plot showing each group's response to the DSQ-PEM questionnaire at both assessment points. Graphing the data will allow readers to visualize how activity tracking plus just-in-time messaging (intervention group) affected each outcome compared to standard care

(control group). For instance, the numerical outcomes can be presented in the form of a graph that illustrates the evolution of each group's mean score at baseline and post. In the context of categorical outcomes, the utilization of a Sankey plot is a recommended approach for the graphical representation of alterations occurring at two timepoints.

We thank the reviewer for this suggestion and have added figures 4 and 5 which reflect the ITT analysis (monthly data points). However, we would like to keep the tables as we believe this is the most appropriate for open science and reproducibility of RCTs. Moreover, any researchers seeking to meta-analyse data from the results of this manuscript will find it easier to use data from the table, rather than a figure.

52. The data sharing statement should include clear instructions on how to contact the corresponding author for reasonable data access requests

The data sharing agreement has been updated:

“Data collected for this study, including individual anonymised participant data and a data dictionary defining each field in the set will be made available to others upon reasonable request to the corresponding author (l.hayse4@lancaster.ac.uk)”

53. **Reviewer #4 (Remarks to the Author):**

Thank you for reviewing our manuscript.

Reviewer #2 (Remarks to the Author):

I would like to express my sincere gratitude for your comprehensive response to the extensive comments. In point 26 of the "Response to Referees Letter", a detailed explanation was provided, describing that the primary activity threshold was determined on the basis of heart rate.

Please accept my apologies for any inconvenience caused; it may be the case that I did not articulate my comment with sufficient precision. The question regarding cycling should pertain to patients with unstable heart rates who were included in the analysis. The heart rate value could not be used as a benchmark in this case. It is my understanding that such activity cannot be recorded and taken into account due to the unstable heart rate. Consequently, the exclusion of these 23 patients from the analysis may be warranted, as their inclusion could potentially influence the results, for instance, if their cycling habits significantly impacted the outcomes.

We thank the reviewer for their suggestion, and we politely disagree these participants should be removed for reasons of equality, bias, and ecological validity, we have included sensitivity analysis where these participants were removed as follows:

“As suggested by a reviewer, we also analysed results without those who were switched to steps rather than heart rate. Repeated measures ANOVA revealed no effect of time ($p=0.213$, $\eta^2p=0.011$; small), group ($p=0.998$, $\eta^2p=0.000$; trivial), or interaction effect ($p=0.776$, $\eta^2p=0.001$; trivial) for the primary outcome variable.”

Reviewer #3 (Remarks to the Author):

We would like to thank the authors for their detailed responses to the points we flagged in our review. Most of the answers are satisfactory.

The intention-to-treat analysis was reported in the Results section, under the subtitle: 'Sensitivity Analysis'. In randomized trials, the purpose of an ITT analysis is to estimate the effect of the intervention while maintaining the benefits of randomization, regardless of adherence to the protocol or any deviations from it. Sensitivity analyses, on the other hand, are additional analyses carried out to check the robustness of the main results to different assumptions. Therefore, the subtitle 'Sensitivity Analysis' must be removed or substituted with 'Intention-to-Treat Analysis'.

We thank the reviewer for this clarity and have deleted this title.

We appreciate the extended and detailed justification of the DSQ-PEM analysis, and we agree with the authors on most points. However, when reviewing the cited papers as examples of studies that have treated the DSQ-PEM as continuous data and analysed it using parametric tests such as ANOVA, we did not find the same approach if we understood correctly. Jason et al. (<https://www.mdpi.com/2035-8377/15/1/1>) and Nepotchatky et al. (<https://www.nature.com/articles/s41598-023-28955-9>) appear to report DSQ results as a 100-point composite score. In contrast, Twomey et al. (<https://pmc.ncbi.nlm.nih.gov/articles/PMC9383197/>) reported DSQ items using the step scoring method, where a frequency and severity score of 2,2 on any items 1–5 is considered indicative of PEM.

We thank the reviewer for their comment. We apologise for being unclear. In our current manuscript, we have also reported DSQ-PEM results as a 100-point composite score (as per Jason et al and Nepotchatky et al). For example, in our results, we communicate “Using the sum of the DSQ-PEM questions 1-5 expressed on a 100-point scale (as per our power calculation), the intervention group value was 48 (95% CI 44-53) at baseline and 46 (95% CI 41-51) post-intervention. The control group value was 47 (95% CI 42-52) at baseline and 44 (95% CI 39-49) at follow-up (interaction effect $p=0.614$, $\eta^2p=0.002$; trivial).”

Additionally, in the last round of revisions reviewer 2 requested “*This paragraph provides an overview of the recovery process for individuals affected by LC. This is also mentioned in the summary in lines 48-49. Could you please clarify whether there are any participants in your group who no longer have symptoms, or whose PEM screening is no longer positive?*” We interpreted this to mean, presentation of data using the step scoring method, as per Twomey et al.. This is a binary yes/no for presentation of PEM as you have correctly identified. To address reviewer 2’s previous comment, we have also presented results in this way as follows:

“To confirm the tests of mean differences above (i.e. ANOVA) were robust, we determine the number of individuals considered positive for PEM (i.e. discrete yes/no). A score of ≥ 2 for both frequency and severity on a specific item (Q1-5) suggests PEM is both present and clinically relevant⁴. Therefore, participants could be either positive or negative for PEM. When groups were combined, McNemar’s Test revealed the number of individuals classified as PEM positive decreased from baseline to follow-up ($p=0.004$). 111 participants were considered PEM positive at baseline and follow-up, 31 participants were PEM positive at baseline but not at follow-up (improvement), 12 participants were PEM negative before but positive after (worsening), and seven participants were PEM negative at both time points. In the intervention group, McNemar’s Test revealed no change in PEM status from baseline to post-intervention ($p=0.394$). 59 participants were considered PEM positive at baseline and post-intervention, 13 participants were PEM positive at baseline but not at post-intervention (improvement), nine participants were PEM negative before but positive after (worsening), and three participants were PEM negative at both time points. In the control group, the number of individuals classified as PEM positive decreased from baseline to follow-up ($p=0.001$). 52 participants were considered PEM positive at baseline and follow-up, 18 participants were PEM positive at baseline but not at follow-up (improvement), three participants were PEM negative before but positive after (worsening), and four participants were PEM negative at both time points. We subsequently used Chi squared test for between group effects which exhibited no differences at baseline ($p=0.411$) or follow-up ($p=0.155$).”

Please let us know if our writing style is unclear or confusing in any way and we can alter the manuscript for clarity.

Reviewer #4 (Remarks to the Author):

Thank you for reviewing the manuscript.